# RDD: Retrieval-Based Demonstration Decomposer for Planner Alignment in Long-Horizon Tasks

**Mingxuan Yan**[1]    **Yuping Wang**[1,2]    **Zechun Liu**[3]    **Jiachen Li**[1*]

[1]University of California, Riverside    [2]University of Michigan    [3]Meta AI
{myan035, yuping.wang, jiachen.li}@ucr.edu   zechunliu@meta.com

## Abstract

To tackle long-horizon tasks, recent hierarchical vision-language-action (VLAs) frameworks employ vision-language model (VLM)-based planners to decompose complex manipulation tasks into simpler sub-tasks that low-level visuomotor policies can handle. Typically, the VLM planner needs finetuning to learn to decompose a new task, which requires target task demonstrations segmented into sub-tasks by either human annotation or heuristic rules. However, without prior knowledge, the heuristic sub-tasks can deviate significantly from the visuomotor policy's training data, thereby degrading task performance. To address these issues, we propose a **R**etrieval-based **D**emonstration **D**ecomposer (**RDD**) that automatically decomposes video demonstrations into sub-tasks with prior by aligning the visual features of the decomposed sub-task intervals with those from the training data of the low-level visuomotor policies. RDD outperforms the state-of-the-art sub-task decomposer on both simulation and real-world tasks, demonstrating robustness across diverse settings. Code and more results are available at rdd-neurips.github.io.

## 1   Introduction

Developing generalist robots that are capable of executing complex, long-horizon tasks in unstructured environments has become one of the central goals of current robotics research. Traditional robotic programming and learning methods often struggle with the variability and complexity inherent in real-world scenarios. Building upon the success of Vision-Language Models (VLMs) and Large Language Models (LLMs), a new class of multi-modal foundation models known as Vision-Language-Action models (VLAs) [1, 2, 3, 4, 5] has emerged specifically for embodied AI applications. As recent studies [6, 7, 8, 9, 10, 11, 12, 13] have shown, integrating high-level planners above the low-level visuomotor policies vastly improves the performance for long-horizon robotic tasks. This has led to the hierarchical VLA paradigm [14, 15, 13, 16, 17, 18, 19, 20]. The planner, often a powerful VLM, performs task planning and reasoning to break down complex tasks into simpler sub-tasks with step-by-step language instructions. A learning-based visuomotor policy, trained on datasets with short-horizon sub-tasks and conditioned on the generated sub-task instructions, performs precise manipulation to complete the sub-tasks one by one, thereby completing long-horizon tasks.

Despite its versatility, a vanilla VLM planner typically needs to be finetuned with human demonstrations when deploying to a given task [18, 14, 16]. To build the dataset for planner finetuning, demonstrations are temporally decomposed to sub-tasks by human annotation [14, 16, 18, 19, 15] or heuristics [13, 15, 21, 22, 23, 24, 25]. However, these methods are neither scalable nor efficient, and, most importantly, they could generate sub-tasks that deviate significantly from the training data of the low-level visuomotor policy. Figure 1 illustrates this dilemma. The state-of-the-art sub-task decomposer UVD [25], which uses a heuristic decomposition rule based on visual feature change-point

---

[*]Corresponding author

39th Conference on Neural Information Processing Systems (NeurIPS 2025).

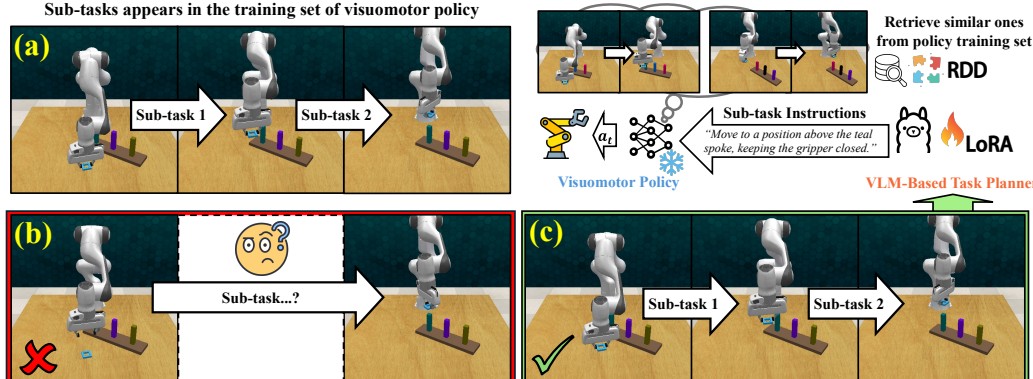

**Figure 1:** The core idea of RDD. **(a)** Two sub-tasks appear in the visuomotor policy's training set, on which the policy has been optimized. **(b)** Existing sub-task decomposers, such as UVD [25], use heuristic decomposition rules and may generate "unfamiliar" sub-tasks that are difficult to handle for the low-level visuomotor policy. **(c)** In contrast, RDD decomposes the demonstration into sub-tasks that are visually similar to the ones in the training set of the visuomotor policy. The sub-tasks are then used to finetune the high-level planner, which gives sub-task instructions to the low-level visuomotor policy and guides it to finish the task step-by-step.

detection, generates sub-tasks that significantly deviate from the training data of the visuomotor policy. Finetuning the planner with these sub-tasks could make the planner generate sub-task instructions that the visuomotor policy is not optimized for, leading to compromised performance.

This gap motivates us to develop an automatic, training-free, and computationally efficient approach that *identifies sub-tasks from a video demonstration with prior*, i.e., the decomposed sub-tasks should be aligned with the training data of the low-level visuomotor policies. To achieve this, we propose a **R**etrieval-based **D**emonstration **D**ecomposer (**RDD**) that decomposes the demonstration into sub-tasks visually similar to the ones in the training set of the visuomotor policy, as illustrated in Figure 1 (c). Inspired by previous work [25], we employ existing visual encoders [26, 27, 28, 29, 30] that encode images into a compact latent space where distance metrics (e.g., angular distance) are effective in describing the semantic relationship between images. To align the sub-tasks to the training data of the low-level visuomotor policy, we build a sub-task visual feature vector database with the visuomotor training set and design an effective sub-task similarity measure to ensure similar sub-task samples can be efficiently retrieved. We formulate sub-task identification as an optimal partitioning problem and employ a dynamic programming-based solver to optimize the sub-task partitioning strategy efficiently. The experiments show that RDD consistently outperforms state-of-the-art methods on both simulation and real-world benchmarks.

The main contributions of this paper are as follows:

- This work is the first to coordinate the high-level planner and low-level visuomotor policy in the hierarchical VLA framework by generating the planner's finetuning dataset that is well aligned with the visuomotor policy to improve the long-horizon task performance.
- We propose RDD, a retrieval-based video sub-task identification algorithm with sub-task prior. Specifically, we model sub-task identification as an optimal partitioning problem, which can be solved efficiently with a dynamic programming solver.
- We evaluate RDD on both simulation and real-world benchmarks. Experimental results show that RDD outperforms the state-of-the-art heuristic decomposer and is robust across various settings.

## 2 Related Work

**Hierarchical VLAs.** While single-stage VLAs [1, 2, 3, 4, 5] achieve promising performance in short-horizon manipulation tasks, long-horizon tasks need an in-depth understanding of the task and general planning ability, which is hard to handle by a single-stage model. To this end, hierarchical structures have emerged as a compelling solution for long-horizon manipulation tasks [14, 15, 13, 16, 17, 18, 19, 20, 10]. As representative examples, Hi Robot [14] and $\pi_{0.5}$ [18] enhance their previous work on visuomotor policy [4, 3] with a VLM-based planner. According to image observation and the overall task goal, the planner provides sub-task instructions at each time step. The low-level

policy, conditioned on the instruction, outputs the final actions. Hierarchical structures also enable error correction and human intervention [13, 16, 14]. However, these methods rely on either human annotation or heuristic rules to identify sub-tasks when finetuning the planner, which is less efficient and could generate sub-tasks that are hard to handle by the visuomotor policy.

**Sub-Task Identification.** Finetuning the high-level planner in hierarchical VLAs requires demonstrations broken down into sub-tasks with associated labels. Manually performing this segmentation [14, 16, 18, 19, 15] is slow and expensive. Human subjectivity also leads to inconsistencies. Heuristic methods [13, 15, 21, 22, 23, 24], such as segmenting based on contact changes or end-effector velocity profiles, require task-specific knowledge for carefully designed rules. In contrast, UVD [25] leverages general visual representation and identifies sub-tasks by detecting frame-by-frame temporal change points of visual embedding distances. However, when applying to hierarchical VLAs, UVD can still sub-optimally decompose sub-tasks, which may deviate significantly from the training data of the visuomotor policy. In contrast, with the sub-task prior, RDD identifies sub-tasks by explicitly aligning the sub-tasks with the training set of the visuomotor policy, enabling seamless coordination between the planner and visuomotor policy.

**Visual Representations.** Considerable efforts have been made to develop visual encoders that embed RGB frames into compact latent vector spaces [26, 27, 28, 29, 30]. Some of these efforts are specially designed for robotics and manipulation scenarios. For instance, R3M [27] uses time-contrastive learning on large datasets of human videos; LIV [26] learns a value function conditioned on both language instructions and images. These visual representations are designed to capture meaningful information about the scene, objects, and potentially their relationships or temporal dynamics.

## 3 Retrieval-Based Demonstration Decomposer (RDD)

### 3.1 Problem Statement

**Visuomotor-Planner Dataset Alignment.** Hierarchical VLAs typically follow an imitation learning framework that trains a low-level visuomotor policy $\pi_\theta(a_t|s_t, o_t, l_t, L)$ and a high-level planner $p_\phi(l_t|s_t, o_t, l_{t-1}, L)$. The latter is usually a VLM. $a_t$ denotes the waypoint action at timestep $t$, including 6-DoF pose and binary gripper state. Both policy $\pi_\theta$ and planner $p_\phi$ are conditioned on the RGB image observation $o_t$, proprioceptive states $s_t$, and the overall task objective description $L$ in natural language, such as "put the cube in the drawer". The policy $\pi_\theta$ is additionally conditioned on a sub-task instruction $l_t$ like "first, pick up the cube", which is determined by the planner $p_\phi$ at time $t$.

During the policy training phase, the raw training dataset $\mathcal{D}^{\text{train}} = \{(\mathcal{S}^i, L^i)\}_{i=1}^{N_{\text{train}}}$ is composed of $N_{\text{train}}$ demonstrations where $\mathcal{S}^i = \{(a_t^i, s_t^i, o_t^i)\}_{t=1}^{T_i}$ and $L^i$ represents the corresponding task objective description. To break the complex long-horizon tasks down to simple instructions required by the low-level policy $\pi_\theta$, a demonstration $\mathcal{S}^i$ is decomposed into a set of partitions $P^i = \{I_j^i\}_{j=1}^{B_i}$ based on task-specific rules or human annotations. The $j$-th interval $\mathcal{I}_j^i = \{\mathcal{S}^i[b_j^i], \ldots, \mathcal{S}^i[e_j^i]\}$ $(b_j^i < e_j^i)$ corresponds to a single coherent sub-task, where $b_j^i, e_j^i$ are indexes of the starting and ending frames. All time steps $t$ within the same interval share the same sub-task instruction $l_t^i = f_{\text{lang}}(\text{prompt}_j)$ labeled manually or generated by a powerful language model. As such, the demonstration is augmented with language descriptions $l_t^i$ to $\mathcal{S}_{\text{aug}}^i = \{(a_t^i, s_t^i, o_t^i, l_t^i)\}_{t=1}^{T_i}$ and the augmented training set is denoted as $\mathcal{D}_{\text{aug}}^{\text{train}} = \{(\mathcal{S}_{\text{aug}}^i, L^i)\}_{i=1}^{N_{\text{train}}}$. The policy $\pi_\theta(a_t|s_t, o_t, l_t, L)$ is then optimized on $\mathcal{D}_{\text{aug}}^{\text{train}}$.

During the high-level planner finetuning phase, given $M$ demonstrations ($M \ll N_{\text{train}}$) for each task, we construct a planner finetuning dataset $\mathcal{D}^{\text{demo}} = \{(\mathcal{S}^i, L^i)\}_{i=1}^M$ and predict the sub-task partitioning strategy $P \in \Pi(\mathcal{S}^i)$ for $\mathcal{S}^i$, where $\Pi(\mathcal{S})$ denotes all possible partitioning over a sequence $\mathcal{S}$:

$$\Pi(\mathcal{S}) = \left\{ P = \{\mathcal{I}_1, \mathcal{I}_2, \ldots, \mathcal{I}_K\} \ \middle| \ \bigcup_{i=1}^K \mathcal{I}_i = \mathcal{S}, \ \mathcal{I}_i \cap \mathcal{I}_j = \emptyset \text{ for } i \neq j \right\}.$$

**Sub-task Identification as Optimal Partitioning Problem.** Finding the optimal sub-task partitioning strategy can be formulated as an optimal partitioning problem, as illustrated in Figure 2:

$$P^{i*} = \arg\max_{P \in \Pi(\mathcal{S}^i)} \mathrm{J}(P), \tag{3.1}$$

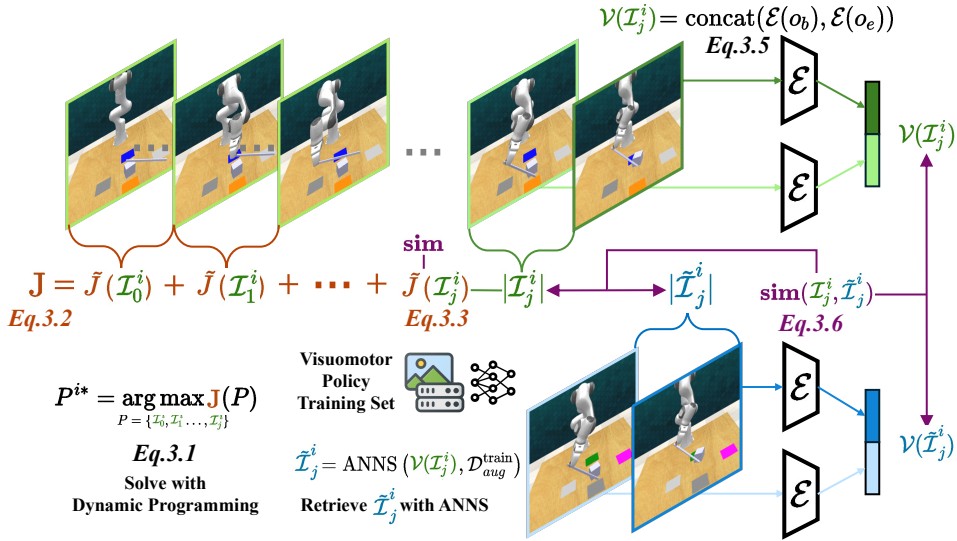

**Figure 2:** RDD formulates sub-task identification as an optimal partitioning problem. Intervals colored in green are proposed segments of the demonstration $\mathcal{S}^i$, and ones colored in blue are retrieved from the visuomotor policy's training set $\mathcal{D}_{aug}^{train}$.

where $J(P)$ is the partitioning strategy scoring function defined on $P$ that evaluates how close the strategy is to the low-level visuomotor policy's training dataset $\mathcal{D}_{aug}^{train}$. Given the partitioning found, $\mathcal{D}^{demo}$ is augmented by $f_{lang}$ and arranged to $\mathcal{D}_{aug}^{demo}$ following the same procedure as $\mathcal{D}_{aug}^{train}$. A pre-trained planner $p_\phi(l_t|s_t, o_t, l_{t-1}, L)$ is then finetuned on $\mathcal{D}_{aug}^{demo}$ with supervised learning to learn to decompose the new task.

## 3.2 Dynamic Programming Solver

Brute-force search of $P^{i*}$ requires $O(2^{N-1})$ times of evaluation of J for a $N$ frame's demonstration, which is computationally intractable. Fortunately, [31] show that when J is interval-wise additive (as illustrated in Figure 2), i.e:

$$J(P) = \sum_{\mathcal{I} \in P} \tilde{J}(\mathcal{I}), \qquad (3.2)$$

which implies $J(P) = J(P_1) + J(P_2)$, $(P_1, P_2) \in \{(P_1, P_2) \,|\, P = P_1 \cup P_2,\ P_1 \cap P_2 = \emptyset\}$, where $\tilde{J}$ is the scoring function of a single interval. The following optimality holds:

**Theorem 3.1** (Principle of Optimality [31]). *Given an additive scoring function* J, *any subset $P'$ of an optimal partition $P^*$ is the optimal partitioning strategy of the intervals it covers.*

This implies that if we find the partial optimal partitioning strategy for $\mathcal{S}^i[0 : j]$, it must be a subset of the global optimal $P^{i*}$. This optimality structure allows a dynamic programming algorithm [31] to find the optimal partition with $O(N^2)$ evaluations of the interval scoring function $\tilde{J}$.

In real-world robot learning scenarios, the duration of a sub-task is limited (typically tens of seconds) [32, 33, 34, 14], thus the complexity of the algorithm can be further improved by ignoring intervals excessively long. We show that: *if the length of the interval is bounded, the complexity can be further reduced to $O(N)$*. We provide the algorithm implementation in Appendix A.1, Algorithm 1, and draw the following conclusion:

**Corollary 3.1.1.** *If the length of every interval is in the range $[L_{min}, L_{max}]$, $0 < L_{min} < L_{max} \le N$, Algorithm 1 finds the optimum with $O\left((L_{max} - L_{min}) \cdot \max(L_{max} - L_{min}, N - L_{max})\right)$ evaluations of the interval scoring function $\tilde{J}$.*

We defer the proof to Appendix A.2. When the maximum sub-task interval length $L_{max}$ is bounded, which is common in robotics learning scenarios, a linear complexity $O(N)$ is achieved. Considering general cases, in this work, we make no assumption on $L_{max}$ and only mildly assume $L_{min} = 2$ for

sanity (a valid interval must have both the starting and ending frame). We additionally remark that Algorithm 1 supports parallel evaluation of the scoring function, as the intervals to be evaluated are determined at the beginning.

**Interval Scoring Function.** Recall that $\tilde{J}$ should reflect how well the proposed interval aligns with the intervals in the training set $\mathcal{D}_{\text{aug}}^{\text{train}}$, we define the interval scoring function $\tilde{J}$ as:

**Definition 3.1.** *The scoring function $\tilde{J}$ for an interval $\mathcal{I}$ is defined as:*

$$\tilde{J}(\mathcal{I}_j^i) = |\mathcal{I}_j^i|\mathbf{sim}\left(\mathcal{I}_j^i, \ \text{ANNS}(\mathcal{V}(\mathcal{I}_j^i), \mathcal{D}_{aug}^{train})\right) = |\mathcal{I}_j^i|\mathbf{sim}(\mathcal{I}_j^i, \tilde{\mathcal{I}}_j^i), \tag{3.3}$$

*where $\mathcal{V}$ maps interval $\mathcal{I}$ into a $d$-dimensional vector representation, $|\mathcal{I}|$ is the duration of the $\mathcal{I}$, and $\text{ANNS}(\mathcal{I}, \mathcal{D}_{aug}^{train})$ represents the approximate nearest neighbor of the interval proposal $\mathcal{I}$ in the training set $\mathcal{D}_{aug}^{train}$ under some distance metric $\delta$ in $\mathbb{R}^d$. $\mathbf{sim}$ is an interval similarity measure. For simplicity, we denote the result of approximate nearest neighbor search for $\mathcal{I}_j^i$ as $\tilde{\mathcal{I}}_j^i$.*

Eq. 3.3 essentially *evaluates how close the proposed interval is to the training set of the visuomotor policy in the training set $\mathcal{D}^{train}$*. Moreover, Def. 3.1 ensures the following notable property:

**Proposition 3.1.** *Suppose an interval $\mathcal{I}_j^i$ can be split into $K$ consecutive parts $\{\mathcal{I}_{j1}^i, \mathcal{I}_{j2}^i, \ldots, \mathcal{I}_{jK}^i\}$, all of which have the same training set similarity score, i.e., $\mathbf{sim}(\mathcal{I}_j^i, \tilde{\mathcal{I}}_j^i) = \mathbf{sim}(\mathcal{I}_{j1}^i, \tilde{\mathcal{I}}_{j1}^i) = \cdots = \mathbf{sim}(\mathcal{I}_{jK}^i, \tilde{\mathcal{I}}_{jK}^i)$. Given the interval scoring function $\tilde{J}$ of Eq. 3.3, and an additive $J$, the following equality holds:*

$$J(\{\mathcal{I}_j^i\}) = J(\{\mathcal{I}_{j1}^i, \mathcal{I}_{j2}^i, \ldots, \mathcal{I}_{jK}^i\}). \tag{3.4}$$

The proof is in Appendix B. This equality implies that $J$ is *ignorant of the number of intervals* when evaluating nested partitionings with the same similarity score. An alternative way to interpret is that, in Eq. 3.3, $\mathbf{sim}$ assigns scores to the sub-task assignment of each timestamp in an interval instead of assigning to the interval as a whole, thus the score summation is irrelevant to the number of intervals in the partitioning strategy.

### 3.3 Interval Similarity and Overall Objective

**Interval Similarity Measures.** As introduced in Section 2, one can embed the RGB image observation $o_t^i$ into a compact latent vector space for similarity measures. We define $\mathcal{V}$ as:

$$\mathcal{V}(\mathcal{I}) = \text{concat}\left(\mathcal{E}(o_b), \mathcal{E}(o_e)\right). \tag{3.5}$$

As illustrated in Figure 2, $o_b, o_e$ are image observations at the beginning and end of $\mathcal{I}$, and $\mathcal{E}$ is the embedding function. This formulation is inspired by former studies [26, 35, 25] that the ending frame (i.e., the goal frame) contains rich information about the sub-task goal and thus can be a distinguishable representation. Eq. 3.5 also includes the starting frame, which is essentially the goal state of the previous sub-task, to aggregate context-related information into the vector representation.

Let the approximate nearest neighbor of $\mathcal{I}_j^i$ be $\tilde{\mathcal{I}}_j^i = \text{ANNS}(\mathcal{I}_j^i, \mathcal{D}_{\text{aug}}^{\text{train}})$ we define the similarity measure $\mathbf{sim}$ between $\mathcal{I}_j^i, \tilde{\mathcal{I}}_j^i$ as:

$$\mathbf{sim}(\mathcal{I}_j^i, \tilde{\mathcal{I}}_j^i) = -\left[\delta(\mathcal{V}(\mathcal{I}_j^i), \mathcal{V}(\tilde{\mathcal{I}}_j^i)) + \alpha\left|1 - \frac{|\mathcal{I}_j^i|}{|\tilde{\mathcal{I}}_j^i|}\right|\right], \tag{3.6}$$

where the first term is the distance between the vector representations of $\mathcal{I}_j^i$ and $\tilde{\mathcal{I}}_j^i$; the second evaluates the relative difference between the temporal durations of two intervals. $\alpha$ is a hyperparameter that controls the weights between temporal and visual similarity.

**Considering OOD Sub-tasks.** While the primary objective of RDD is to align the planner with the visuomotor policy's existing capabilities, in real-world applications, out-of-distribution (OOD) sub-tasks not learned by the low-level visuomotor may exist. In such scenarios, the objective changes to: *aligning sub-task intervals to both existing visuomotor sub-tasks and general sub-tasks*, and the newly identified sub-tasks will be used to finetune both the visuomotor and the planner. Firstly, to detect the existence of new sub-tasks in demonstrations, one can quantify the novelty of a demonstration by $\Delta = \frac{1}{|P|}\sum_{\mathcal{I} \in P} \tilde{J}(\mathcal{I})$, the average similarity score of the optimal partition $P$ found by the standard

RDD algorithm. A low value of $\Delta$ indicates a low averaged similarity, which signals novel sub-tasks. An alternate interval similarity measure **sim** for the OOD setting is defined:

$$\mathbf{sim}(\mathcal{I}_j^i, \tilde{\mathcal{I}}_j^i) = \underbrace{-\delta(\mathcal{V}_e(\mathcal{I}_j^i), \mathcal{V}_e(\tilde{\mathcal{I}}_j^i))}_{\text{retrieval}} + \underbrace{\beta G(\mathcal{I}_j^i)}_{\text{general}}, \tag{3.7}$$

where $\mathcal{V}_e(\mathcal{I}) = \mathcal{E}(o_e)$ and only the ending frame is used to calculate the semantic distance due to unpredictable OOD sub-task durations; G evaluates how well a proposed interval aligns with "general" sub-tasks. The hyperparameter $\beta$ balances the trade-off between aligning with visuomotor sub-tasks and discovering novel, generalizable sub-tasks. G can be implemented using heuristic general sub-task identification functions like UVD [25] to measure how well an interval conforms to generic change-point detection heuristics:

$$G(\mathcal{I}) = -\frac{1}{|\mathcal{I}|}\mathbf{abs}(b - \mathbf{UVD}(e, \mathcal{I})), \tag{3.8}$$

where $b, e$ represent the index of the beginning and ending frame of interval $\mathcal{I}$. $\mathbf{UVD}(e, \mathcal{I})$ gives the index of the UVD predicted beginning frame, given the goal frame on $e$.

**Approximate Nearest Neighbor Search.** Considering the vast number of intervals in $\mathcal{D}_{\text{aug}}^{\text{train}}$ and the high-dimensional vector space, we adopt approximate nearest neighbor search (ANNS) to implement the nearest neighbor searcher. We choose the popular random-projection-trees-based method Annoy [36] as the ANNS implementation, which is computationally efficient and shows good robustness on various data [37]. RDD can also work with GPU-accelerated ANNS libraries like FAISS [38] for further acceleration.

**Overall Optimization Objective.** By substituting Eq. 3.2 and Eq. 3.3 into Eq. 3.1, we have the complete definition of the optimization problem as:

$$P^{i*} = \operatorname*{arg\,max}_{P \in \Pi(\mathcal{S}^i)} \sum_{\mathcal{I}_j^i \in P} |\mathcal{I}_j^i|\mathbf{sim}(\mathcal{I}_j^i, \tilde{\mathcal{I}}_j^i), \tag{3.9}$$

where **sim** is defined by Eq. 3.6 or alternatively Eq. 3.7 for OOD settings. The optimal partitioning strategy $P^{i*}$ of demonstration $\mathcal{S}^i$ can be solved by Algorithm 1.

## 4 Experiments

**Implementation and Parameter Settings.** We adopt RACER [13] as the base hierarchical VLA framework, which uses RVT [39] as the low-level visuomotor policy $\pi_\theta$ and the recent LLaVa-based VLM llama3-llava-next-8B [40] as the pre-trained base model for planner $p_\phi$. We use the pre-trained RVT policy $\pi_\theta$ provided by RACER [13] trained $\mathcal{D}_{\text{aug}}^{\text{train}}$ and the validation set of RLBench (labeled with the same decomposition rule as in $\mathcal{D}_{\text{aug}}^{\text{train}}$). During the deployment phase, the planner is finetuned for two epochs on $\mathcal{D}_{\text{aug}}^{\text{demo}}$ using LoRA [41], with the rank of 128 and a scaling factor of 256 following RACER. The finetuning process takes about 5 minutes with 4 NVIDIA 6000 Ada GPUs. For base parameter settings, we set the weighting factor $\alpha = 1$ and interval similarity measure **sim** in Eq. 3.6 for non-OOD scenarios, and use LIV [26] as the visual encoder $\mathcal{E}$ that is specifically designed for manipulation tasks. We use Gemini-1.5-flash [42] to generate sub-task language instructions for proposed intervals in $\mathcal{D}_{\text{aug}}^{\text{demo}}$.

**Visuomotor Policy Training Dataset and Vector Database.** We evaluate RDD on the RLBench [32] robot manipulation benchmark. The visuomotor policy training set $\mathcal{D}_{\text{aug}}^{\text{train}}$ is adapted from [13]. $\mathcal{D}^{\text{train}}$ originally consists of 1908 teleoperated demonstrations from the RLBench's training set. When generating $\mathcal{D}_{\text{aug}}^{\text{train}}$, RACER additionally augmented it with heuristic failure-recovery samples, resulting in a training dataset with 10,159 demonstrations. In this work, we only use the original 1908 demonstrations to exclude interference. Demonstrations are decomposed into 12700 sub-task intervals using a task-specific heuristic decomposer based on motion and gripper states. Generally, the decomposer will mark a goal state of a sub-task whenever: 1) the gripper state closes or opens, 2) the arm stops for a pre-defined duration, and 3) the end of the demonstration. More details about this heuristic can be found in Section III.B of [13]; RACER uses GPT-4-turbo as the language labeling function $f_{\text{lang}}$ to annotate the sub-task intervals, given the language descriptions of the robot movement and initial environment setup.

Table 1: Multi-task success rates (%) on RLBench.

| Method | Avg. Succ. (↑) | Avg. Rank (↓) | Close Jar | Install Bulb | Meat off Grill | Open Drawer | Place Wine | Push Buttons |
|---|---|---|---|---|---|---|---|---|
| w/o Finetune | $52.6_{\pm 8.2}$ | $4.5_{\pm 1.2}$ | $27.6_{\pm 26.4}$ | $34.8_{\pm 14.2}$ | $46.4_{\pm 26.8}$ | $95.6_{\pm 6.1}$ | $83.2_{\pm 13.0}$ | $54.8_{\pm 9.1}$ |
| Uniform | $71.3_{\pm 5.4}$ | $3.1_{\pm 1.2}$ | $\mathbf{46.4}_{\pm 29.9}$ | $51.2_{\pm 19.2}$ | $76.4_{\pm 22.4}$ | $\mathbf{100.0}_{\pm 0.0}$ | $80.8_{\pm 14.5}$ | $82.0_{\pm 7.8}$ |
| UVD | $71.4_{\pm 5.1}$ | $3.0_{\pm 1.3}$ | $44.0_{\pm 28.7}$ | $\mathbf{54.8}_{\pm 20.0}$ | $\mathbf{85.2}_{\pm 20.6}$ | $\mathbf{100.0}_{\pm 0.0}$ | $80.8_{\pm 15.3}$ | $67.2_{\pm 13.6}$ |
| RDD (Ours) | $\mathbf{74.9}_{\pm 6.9}$ | $\mathbf{2.2}_{\pm 0.9}$ | $46.0_{\pm 28.2}$ | $52.8_{\pm 16.4}$ | $84.4_{\pm 21.1}$ | $99.2_{\pm 2.4}$ | $\mathbf{86.4}_{\pm 15.4}$ | $\mathbf{84.0}_{\pm 7.8}$ |
| *Expert* | $75.1_{\pm 4.7}$ | $2.2_{\pm 1.0}$ | $50.4_{\pm 33.1}$ | $50.4_{\pm 13.3}$ | $94.4_{\pm 9.7}$ | $99.2_{\pm 2.4}$ | $81.6_{\pm 15.0}$ | $85.6_{\pm 6.0}$ |

| Method | Put in Cupboard | Put in Drawer | Put in Safe | Drag Stick | Slide Block | Sweep to Dustpan | Turn Tap |
|---|---|---|---|---|---|---|---|
| w/o Finetune | $\mathbf{41.2}_{\pm 20.1}$ | $36.4_{\pm 28.8}$ | $58.8_{\pm 23.3}$ | $36.0_{\pm 21.8}$ | $57.2_{\pm 14.9}$ | $22.8_{\pm 32.5}$ | $89.2_{\pm 13.4}$ |
| Uniform | $36.8_{\pm 15.4}$ | $\mathbf{98.0}_{\pm 2.7}$ | $92.4_{\pm 10.8}$ | $64.8_{\pm 16.7}$ | $64.4_{\pm 9.9}$ | $34.8_{\pm 37.7}$ | $\mathbf{98.8}_{\pm 3.6}$ |
| UVD | $35.2_{\pm 12.1}$ | $90.4_{\pm 8.6}$ | $96.8_{\pm 6.6}$ | $\mathbf{74.4}_{\pm 29.2}$ | $\mathbf{66.8}_{\pm 21.2}$ | $43.6_{\pm 24.6}$ | $89.6_{\pm 11.1}$ |
| RDD (Ours) | $\mathbf{41.2}_{\pm 17.1}$ | $97.2_{\pm 3.1}$ | $\mathbf{98.4}_{\pm 3.2}$ | $68.0_{\pm 25.0}$ | $65.2_{\pm 14.3}$ | $\mathbf{57.2}_{\pm 29.7}$ | $94.0_{\pm 5.1}$ |
| *Expert* | $39.6_{\pm 15.6}$ | $91.2_{\pm 7.3}$ | $97.6_{\pm 5.1}$ | $75.2_{\pm 24.6}$ | $66.4_{\pm 22.0}$ | $48.8_{\pm 35.5}$ | $96.0_{\pm 5.7}$ |

Given $\mathcal{D}_{\text{aug}}^{\text{train}}$, we build a vector database following Eq. 3.5 and employ Annoy [36] as the ANNS algorithm to retrieve the approximate nearest neighbor. For each frame, to exclude the inference of occlusion, we concatenate the representation vectors of the front-view and gripper-view images into one. We apply the same configuration to UVD for fair comparison. For Annoy, we set the number of random-projection trees to 10 and let the searcher search through all trees at runtime. We empirically find that the choices of the ANNS algorithm or search parameters have a minor impact on the performance. We use angular distance as the distance measure $\delta$, which is written as $\sqrt{2(1 - cos(u, v))}$ for normalized vectors $u, v$. The finetuning dataset $\mathcal{D}_{\text{aug}}^{\text{demo}}$ is built on RLBench's validation set following the same procedure except that the decomposition strategy is replaced by RDD. Each task has three demonstrations.

**Evaluation Metrics and Baselines.** We evaluate the performance of RDD and baselines in terms of multi-task success rates and corresponding rankings across 13 RLBench tasks[2]. We compare our approach with a variety of baselines that adopt different sub-task identification strategies:

- **Expert** [13]: The same expert heuristic decomposer used in $\mathcal{D}_{\text{aug}}^{\text{train}}$ as performance upper bound.
- **UVD** [25]: A task-agnostic decomposer that detects change points of learning-based visual features.
- **Uniform**: A decomposer that divides each demonstration into 10 partitions with equal duration.
- **w/o Finetune**: The planner $p_\phi$ is the pre-trained VLM model without finetuning on $\mathcal{D}^{\text{demo}}$.

### 4.1 Quantitative Results and Analysis

**Multi-Task Performance on RLBench.** Table 1 shows the overall performance of RDD and baseline methods on multiple manipulation tasks using the base setting in Section 4. Results are averaged over 10 random seeds. RDD achieves a *near-oracle performance* and only compromises the success rate of merely $0.2\%$ compared with the expert decomposer, our performance upper bound. On the other hand, we observe that UVD performs similarly to the naive uniform sampling strategy. It implies that the change points of learning-based visual features are not always aligned with the samples in $\mathcal{D}_{\text{aug}}^{\text{train}}$. By aligning the high-level planner to the knowledge of low-level policy, RDD outperforms the baseline methods that blindly decompose the demonstrations without this knowledge. It also suggests that finetuning is necessary for VLM-based planners. All finetuning-based methods achieve over $35\%$ improvement over the vanilla Llama model.

**Choice of Visual Representation.** As an important building block of RDD, the choice of visual representation is of great importance. Table 2 shows the performance of RDD when adopting different visual encoders $\mathcal{E}$, including robotics specialized encoders: LIV [26], R3M [27], VIP [35], VC-1 [28]; and encoders for general vision tasks: CLIP [43], DINOv2 [29] and ResNet [44] pre-trained for ImageNet-1k classification. Results are averaged over three random seeds.

---

[2] Tasks on which the low-level visuomotor policy has a decent performance (success rate $> 35\%$ with expert planner). It excludes the interference of poorly optimized visuomotor when evaluating planners. Performance on all 18 tasks can be found in Appendix C.

It can be seen that *RDD shows good robustness with various visual encoders* and consistently outperforms baselines with the majority of encoders except for VC-1 and VIP, which demonstrates the strong robustness of RDD. VC-1 and VIP, on the other hand, are the only models that do not involve any form of language integration during training and perform the worst among all encoders. *This implies the importance of language integration for visual encoders in VLA perception for semantic information retrieval.* For instance, subtle pixel differences, such as the change of gripper state, may have a significant difference in language description. Surprisingly, ResNet, whose training does not explicitly involve language supervision, demonstrates a strong performance. The reason may be that its training dataset, ImageNet-1k, implicitly correlates its latent space with the language image labels.

**Weighting Parameters.** Table 3 shows the impact of $\alpha$ on the performance of RDD. Results are averaged over three random seeds. When $\alpha = 0$, i.e., there is no temporal alignment, and the algorithm is confused about sub-tasks whose beginning and ending frames are similar (e.g., reciprocating motion). On the other hand, overly relying on the temporal similarity ignores the semantic relationship between intervals and leads to performance degradation. We also evaluate the impact of the $\beta$ in Table 5 for OOD scenarios, and the result shows that RDD is less sensitive to $\beta$. The choice of $\beta$ depends on specific applications and user needs.

**Number of Demonstrations in $\mathcal{D}^{\mathbf{demo}}$.** To explore the data efficiency of RDD, Table 4 shows its averaged success rates under different numbers of demonstrations in $\mathcal{D}^{\mathrm{demo}}_{\mathrm{aug}}$. Results are averaged over three random seeds. Specifically, we break the three-demonstration base setting dataset into three non-overlapping datasets with one demonstration per task to avoid bias induced by varying demonstration qualities. This result shows a high data efficiency of RDD. We credit this efficiency to the less-noisy keyframes provided by RDD, which are more informative for VLM to learn the underlying decomposition rules.

**Performance on Real-world and OOD sub-tasks.** Here we demonstrate RDD's performance on both real-world and settings where the OOD sub-task appears. We first evaluate RDD on the real-world manipulation benchmark AgiBotWorld-Alpha [33]. We test RDD and UVD on the "supermarket" task, using 152 demos to build the RDD database and 37 demos for testing. For OOD sub-tasks, we test RDD on the human-operated demonstration dataset from RoboCerebra [34], which features highly diverse demonstrations in terms of objects, task goals, and arrangements. We use 560 demos to build the RDD database and test on the remaining 140 demos. We use the similarity measure **sim** in Eq. 3.7 for the OOD setting.

**Table 2:** Results when using different visual encoders $\mathcal{E}$. Full results on all tasks can be found in Table 9 in the appendix.

| Visu. Repr. | Avg. Succ. ($\uparrow$) | Avg. Rank ($\downarrow$) |
|---|---|---|
| LIV | 81.1 $\pm$ 0.9 | 3.7 $\pm$ 1.6 |
| R3M | 80.0 $\pm$ 3.5 | 3.9 $\pm$ 1.7 |
| VIP | 75.3 $\pm$ 3.4 | 4.1 $\pm$ 2.0 |
| VC-1 | 75.5 $\pm$ 3.1 | 3.8 $\pm$ 2.2 |
| CLIP | 78.2 $\pm$ 2.1 | 4.7 $\pm$ 2.0 |
| DINOv2 | 78.4 $\pm$ 2.4 | 4.5 $\pm$ 1.8 |
| ResNet | **81.1** $\pm$ 2.5 | **3.4** $\pm$ 1.5 |

**Table 3:** Results when tuning the weighting parameter $\alpha$. Full results on all tasks can be found in Table 10.

| $\alpha$ | Avg. Succ. | Avg. Rank |
|---|---|---|
| 0 | 75.0 $\pm$ 2.5 | 3.0 $\pm$ 1.0 |
| 0.5 | 75.7 $\pm$ 2.4 | 2.5 $\pm$ 0.7 |
| 1 | **81.1** $\pm$ 0.9 | 2.3 $\pm$ 1.4 |
| 2 | 76.2 $\pm$ 3.0 | **2.2** $\pm$ 0.8 |

**Table 4:** Results with different numbers of demonstrations per task in $\mathcal{D}^{\mathrm{demo}}_{\mathrm{aug}}$. Full results on all tasks are in Table 11.

| Demo. Num. | Avg. Succ. ($\uparrow$) | Avg. Rank ($\downarrow$) |
|---|---|---|
| 1 (RDD) | 77.9 $\pm$ 4.5 | 2.0 $\pm$ 0.9 |
| 3 (RDD) | **81.1** $\pm$ 0.9 | **1.6** $\pm$ 0.6 |
| 3 (UVD) | 75.6 $\pm$ 1.8 | 2.4 $\pm$ 0.6 |

**Table 5:** Performance on real-world and OOD sub-tasks (IoU).

| Method | AgiBot. (Real World) | LIBERO (OOD) |
|---|---|---|
| UVD | 0.506 | 0.598 |
| RDD | **0.706** | / |
| RDD ($\beta = 0.25$) | / | 0.624 |
| RDD ($\beta = 0.10$) | / | **0.630** |
| RDD ($\beta = 0.05$) | / | 0.614 |

We evaluated the quality of the decomposition against ground-truth segmentations using the mean intersection over union (mIoU). As shown in Table 5, RDD outperforms UVD on real-world data. Under OOD settings, RDD consistently outperforms UVD by leveraging potential similarity between sub-tasks.

**Speed and Scalability.** We test the running time of Algorithm 1 with different numbers of frames on AMD EPYC 9254 using **one** CPU core. Figure 3 plots the running time with/without the prior knowledge of the maximum length of interval $L_{\max}$. The results show that the complexity with $L_{\max}$ grows linearly with the number of frames, which aligns with our conclusion in Corollary 3.1.1, which indicates that when $L_{\max}$ is determined, the complexity of Algorithm 1 will be $O(N)$.

Note that *Algorithm 1 supports parallel evaluation of the scoring function $\tilde{J}$, and the latency can be significantly reduced with multi-processing. Also, we demonstrate the scalability of RDD when working with GPU-accelerated ANNS algorithms like FAISS [38] in Appendix D.

**Necessity of Finetuning on Target Tasks:** One may ask if the planner can transfer to an unseen new task in zero-shot. We thus build a new planner finetuned before deployment on the training set of the following tasks: "Close Jar", "Insert Peg", and "Install Bulb" as the baseline, which learns the visual features but not the task decompositions. Then, we test its performance on the remaining tasks. Results in Table 6 are averaged across 10 random seeds, and we also exclude tasks where the visuomotor fails. The results prove the necessity of fine-tuning on target tasks.

**Decompose with VLMs:** VLMs pretrained on internet-scale data are promising to process a variety of video understanding tasks. In Table 7, we compared RDD with a Gemini-2.5-pro [42]-based decomposer with the following prompt:

*There is a robot doing a task, which can be segmented into multiple steps. A keyframe is where the robot finishes the previous step and begins the next. Can you help me find ALL indexes of keyframes? Please return a list of indices, for example: [15, 65, 105, ...]. Note that the frame index starts from 0 instead of 1.*

As shown, RDD outperforms Gemini-2.5-pro despite its powerful general video understanding abilities. This result highlights the necessity of the planner aligning and the effectiveness of RDD.

**Extended Evaluations and Discussions.** We provide extended evaluations results in C and further discussions in Appendix D. We also provide a conceptual speed evaluation of RDD when working with the GPU-accelerated ANNS method FAISS [38] in Appendix D.1.

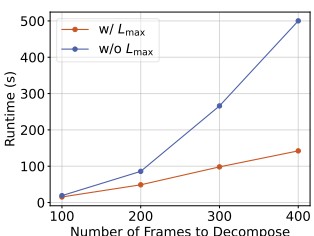

**Figure 3:** Linear scaling of running time of Algorithm 1 with $L_{max}$. Tested with one CPU core.

**Table 6:** Vanilla Planner without finetuning on the target task. Full results on all tasks are in Table 12.

| Method | Avg. Succ. (↑) | Avg. Rank (↓) |
|---|---|---|
| w/o finetuning on target task | 77.9 ±4.3 | 1.6 ±0.5 |
| RDD (Ours) | **79.6** ±7.2 | **1.4** ±0.5 |

**Table 7:** Comparing RDD with Gemini-2.5-pro. Full results on all tasks are in Table 13.

| Method | Avg. Succ. (↑) | Avg. Rank (↓) |
|---|---|---|
| Gemini-2.5-pro | 72.6 ±4.7 | 1.7 ±0.4 |
| RDD (Ours) | **74.9** ±6.9 | **1.3** ±0.4 |

### 4.2 Qualitative Results and Analysis

Figure 4 visualizes the decomposition results of RDD and UVD on both real-world and simulation benchmarks. We can observe that RDD is robust to task-irrelevant interference and can robustly identify the sub-tasks that are close to the expert sub-task division. Also, RDD demonstrates strong robustness to nuanced arm movements, where the keyframe localization is challenging precisely. Conversely, UVD fails to locate keyframes precisely, and the generated sub-tasks largely deviate from expert sub-tasks.

## 5 Discussions and Future Works

**Visuomotor Training Data Generation based on Source Dataset**: While this work applies RDD to planner-visuomotor alignment, it can also be used to generate additional sub-task training data for visuomotor *aligned with a labeled source dataset*. By aligning the sub-task interval visual features with the existing source dataset, RDD may make the newly labeled data easier to learn, allowing the visuomotor reuse learned knowledge from the source dataset.

**Specific Sub-task Interval Features**: RDD measures sub-task interval similarity in the single-frame image feature space. Some applications, such as hierarchical vision-language navigation [19], which require the planner to use historical landmark images, may necessitate specialized designs of the similarity score function.

**Data Quality of the Source Dataset and Data Curation**: As a retrieval-based sub-task identification method, RDD's primary objective is to let the high-level planner effectively utilize the skills that the low-level visuomotor policy *already possesses*. Therefore, RDD is agnostic to the "optimality" of the

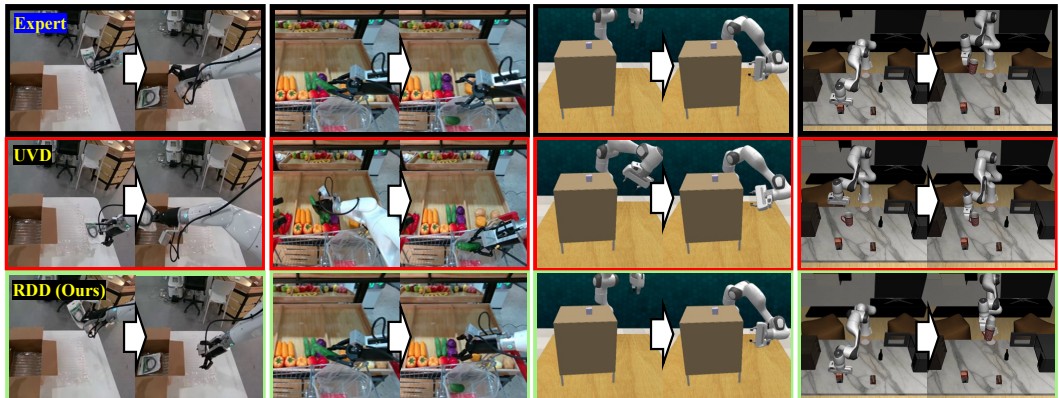

**Figure 4:** Qualitative results of RDD and UVD functioning on both real-world (AgiBotWorld) and simulation (RLBench and LIBERO) benchmarks. Blocks outlined in black are sub-tasks decomposed by the same task-specific heuristic used in the visuomotor policy's training set; blocks outlined in green are sub-tasks found by RDD; and blocks outlined in red are sub-tasks found by UVD.

skills themselves. This ensures the planner generates commands that the policy can reliably execute, rather than potentially "better" ones it cannot handle.

On the other hand, in scenarios where the visuomotor policy's training data contains significant noisy samples that the policy fails to learn, RDD can be easily integrated with dataset curation techniques [45, 46]. These methods can serve as a pre-processing step to filter the visuomotor training set. For instance, CUPID [45] computes an "action influence" score for state-action pairs that can be used to evaluate each segment's contribution to the policy's final behavior. By applying a simple threshold, low-influence or flawed segments can be pruned from the dataset before RDD uses it as a reference. This would prevent catastrophic failures by ensuring RDD aligns demonstrations only with high-quality, influential sub-tasks.

## 6 Conclusion

In this work, we present the Retrieval-based Demonstration Decomposer (RDD), a training-free decomposition method that aligns the high-level task planner and low-level visuomotor policy in hierarchical VLAs. By retrieving and aligning sub-task segments with the low-level policy's training data, RDD enables an effective planner that fully exploits the capability of the visuomotor policy. We formally formulate the sub-task identification task into an optimal partitioning problem, which can be efficiently solved by dynamic programming with our novel sub-task interval scoring function. Experiment results demonstrate that RDD outperforms state-of-the-art demonstration decomposers. RDD offers a scalable and promising solution for sub-task identification, opening new avenues for planner-policy coordination in hierarchical robot learning systems.

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

# Appendix

## A Algorithm Details

### A.1 Dynamic Programming Solver to Problem 3.1

Algorithm 1 shows the dynamic programming solver. $L_{\max}$ and $L_{\min}$ are user-specified parameters that determine the minimum and maximum length of proposed sub-task intervals. $\tilde{J}$ is the interval scoring function.

---

**Algorithm 1** MaxSumPartition

---

**Require:** Sequence $u = [u_1, u_2, \ldots, u_n]$, scoring function $\tilde{J}$, integer $L_{\min}$, integer $L_{\max}$
**Ensure:** Maximum score sum and partition of $u$
 1: Initialize $dp[0 \ldots n] \leftarrow -\infty, parts[0 \ldots n] \leftarrow \emptyset$
 2: $dp[0] \leftarrow 0$
 3: **for** $i = L_{\min} + 1$ to $n$ **do**
 4:     bestScore $\leftarrow -\infty$
 5:     bestPartition $\leftarrow \emptyset$
 6:     **for** $j = 0$ to $i$ **do**
 7:         **if** $L_{\min} \leq i - j \leq L_{\max}$ **then**
 8:             $segment \leftarrow u[j:i]$
 9:             $s \leftarrow \tilde{J}(segment)$                                  ▷ can be evaluated in parallel before loops
10:             **if** $dp[j] + s >$ bestScore **then**
11:                 bestScore $\leftarrow dp[j] + s$
12:                 bestPartition $\leftarrow parts[j] \cup \{segment\}$
13:             **end if**
14:         **end if**
15:     **end for**
16:     **if** bestPartition $\neq \emptyset$ **then**
17:         $dp[i] \leftarrow$ bestScore
18:         $parts[i] \leftarrow$ bestPartition
19:     **else**
20:         $dp[i] \leftarrow dp[i-1]$
21:         $parts[i] \leftarrow parts[i-1]$
22:     **end if**
23: **end for**
24: **return** $(dp[n], parts[n])$

---

### A.2 Proof of Correctness and Complexity of Algorithm 1

*Proof.* The correctness when $L_{\min} = 1, L_{\max} = |\mathcal{S}^i|$ (we denote the algorithm under this case as Algorithm 0) has been proven in *Proof 2* of Jackson et al. [31]. It is sufficient to prove the cases when $1 < L_{\min} < L_{\max} < |\mathcal{S}^i|$. Notice that Algorithm 1 is equivalent to a special case of Algorithm 0 by constructing an adapted scoring function defined as Algorithm 2, where the score of invalid intervals

---

**Algorithm 2** AdaptedScoreFunc

---

**Require:** Sub-sequence $u' = [u_1, u_2, \ldots, u_m]$, scoring function $\tilde{J}$, integer $L_{\min}$, integer $L_{\max}$
**Ensure:** Adapted score of $u'$
 1: **if** $L_{\min} \leq |u'| \leq L_{\max}$ **then**
 2:     **return** $\tilde{J}(u')$
 3: **else**
 4:     **return** $-\infty$
 5: **end if**

---

is $-\infty$. Note that ADAPTEDSCOREFUNC preserves the additiveness of J, because if any interval in a strategy $P$ violates the length assumption, $P$ is also invalid, i.e., $J(P) = -\infty$. Given the facts: 1)

the correctness of Algorithm 0 has been proven by *Proof 2* [31]; 2) Algorithm 1 is equivalent to a special case of Algorithm 0, by implication, the correctness of Algorithm 1 is proved.

As for complexity, let $N$ be the length of the demonstration and $M$ be the number of evaluations to $\tilde{J}$, we have:

$$
\begin{aligned}
M &= \sum_{j=2}^{L_{\max} - L_{\min} + 1} j + (N - L_{\max})(L_{\max} - L_{\min} + 1) \\
&= \frac{(L_{\max} - L_{\min} + 3)(L_{\max} - L_{\min})}{2} + (N - L_{\max})(L_{\max} - L_{\min} + 1) \\
&= O\left((L_{\max} - L_{\min}) \cdot \max(L_{\max} - L_{\min}, N - L_{\max})\right)
\end{aligned}
$$

$\square$

## B  Proof of Proposition 3.1

*Proof.* Let the identical similarity scores equal $s$, and let $b_j^i, e_j^i$ be the starting and ending indexes of interval $\mathcal{I}_j^i$, respectively. By applying Eq. 3.2 and Eq. 3.3 we rewrite the left side of Eq. 3.1 to:

$$
\mathrm{J}(\{\mathcal{I}_j^i\}) = \tilde{J}(\mathcal{I}_j^i) = (e_j^i - b_j^i)s
$$

And the right side:

$$
\begin{aligned}
\mathrm{J}(\{\mathcal{I}_{j1}^i, \mathcal{I}_{j2}^i, \ldots, \mathcal{I}_{jK}^i\}) &= \sum_{k=1}^{K} \tilde{J}(\mathcal{I}_{jk}^i) \\
&= (e_{j1}^i - b_{j1}^i + e_{j2}^i - b_{j2}^i + \cdots + e_{jk}^i - b_{jk}^i)s \\
&= \underbrace{(e_{j1}^i - b_j^i + e_{j2}^i - e_{j1}^i + \cdots + e_j^i - e_{j(k-1)}^i)}_{\text{Since intervals are consecutive.}} s \\
&= (e_j^i - b_j^i)s \\
&= \mathrm{J}(\{\mathcal{I}_j^i\})
\end{aligned}
$$

$\square$

## C  Additional Quantitative Results

Tables 8-13 provide the complete multi-task performances of the results in Section 4.1, including ones where the visuomotor policy fails.

## D  Discussions

### D.1  Work with GPU-accelerated ANNS

The nearest neighbor (NN) search in RDD can be significantly accelerated using GPU-accelerated libraries like FAISS [38]. We conduct experiments on a typical database of 10 million entries (mainstream policy training dataset scale, as shown in Section D.2) of 2048 dimensions (same dimension as our main experiment in Table 1) As shown in Table 14, FAISS can achieve $> 300$ NN queries per second on one NVIDIA 4090 GPU. Under this setting, RDD only needs $< 2$ minutes to decompose a 500-frame video (5 fps), with a max interval length of 100 frames. (44549 NN queries in total). In other words, as part of the offline dataset building process, RDD can decompose demonstrations at a high speed of 4.3 fps, which shows the high scalability of RDD.

### D.2  Scale of Mainstream Robotics Datasets

To support the aforementioned experiment settings, here we provide the scale of some of the most popular open-sourced robotics datasets. In summary, assuming each demonstration can be

**Table 8:** Main results with all RLBench Tasks.

| Method | Avg. Succ. (↑) | Avg. Rank (↓) | Close Jar | Insert Peg | Install Bulb | Meat off Grill | Open Drawer | Place Cups | Sort Shape | Place Wine |
|---|---|---|---|---|---|---|---|---|---|---|
| w/o Finetune | 39.7 ±6.5 | 4.3 ±1.3 | 27.6 ±26.4 | 5.6 ±6.7 | 34.8 ±14.2 | 46.4 ±26.8 | 95.6 ±6.1 | 3.2 ±4.3 | 16.0 ±11.7 | 83.2 ±13.0 |
| Uniform | 54.5 ±4.1 | 3.1 ±1.2 | 46.4 ±29.9 | 8.8 ±11.8 | 51.2 ±19.2 | 76.4 ±22.4 | 100.0 ±0.0 | 0.8 ±1.6 | 25.6 ±9.2 | 80.8 ±14.5 |
| UVD [25] | 54.3 ±3.9 | 3.2 ±1.2 | 44.0 ±28.7 | 10.4 ±14.1 | 54.8 ±20.0 | 85.2 ±20.6 | 100.0 ±0.0 | 1.2 ±1.8 | 25.2 ±11.0 | 80.8 ±15.3 |
| Expert [13] | 57.6 ±3.3 | 2.0 ±0.9 | 50.4 ±33.1 | 12.0 ±17.9 | 50.4 ±13.3 | 94.4 ±9.7 | 99.2 ±2.4 | 3.2 ±3.9 | 26.0 ±10.6 | 81.6 ±15.0 |
| RDD (Ours) | 57.3 ±5.3 | 2.4 ±1.1 | 46.0 ±28.2 | 16.8 ±18.6 | 52.8 ±16.4 | 84.4 ±21.1 | 99.2 ±2.4 | 2.0 ±2.0 | 32.4 ±10.2 | 86.4 ±15.4 |

| Method | Push Buttons | Put in Cupboard | Put in Drawer | Put in Safe | Drag Stick | Slide Block | Stack Blocks | Stack Cups | Sweep to Dustpan | Turn Tap |
|---|---|---|---|---|---|---|---|---|---|---|
| w/o Finetune | 54.8 ±9.1 | 41.2 ±20.1 | 36.4 ±28.8 | 58.8 ±23.3 | 36.0 ±21.8 | 57.2 ±14.9 | 2.8 ±2.6 | 2.8 ±3.6 | 22.8 ±32.5 | 89.2 ±13.4 |
| Uniform | 82.0 ±7.8 | 36.8 ±15.4 | 98.0 ±2.7 | 92.4 ±10.8 | 64.8 ±16.7 | 64.4 ±9.9 | 13.6 ±7.8 | 5.2 ±4.7 | 34.8 ±37.7 | 98.8 ±3.6 |
| UVD [25] | 67.2 ±13.6 | 35.2 ±12.1 | 90.4 ±8.6 | 96.8 ±6.6 | 74.4 ±29.2 | 66.8 ±21.2 | 9.6 ±6.2 | 1.6 ±3.7 | 43.6 ±24.6 | 89.6 ±11.1 |
| Expert [13] | 85.6 ±6.0 | 39.6 ±15.6 | 91.2 ±7.3 | 97.6 ±5.1 | 75.2 ±24.6 | 66.4 ±22.0 | 14.8 ±11.2 | 5.2 ±4.4 | 48.8 ±35.5 | 96.0 ±5.7 |
| RDD (Ours) | 84.0 ±7.8 | 41.2 ±17.1 | 97.2 ±3.1 | 98.4 ±3.2 | 68.0 ±25.0 | 65.2 ±14.3 | 5.2 ±3.6 | 1.6 ±2.7 | 57.2 ±29.7 | 94.0 ±5.1 |

**Table 9:** Multi-task performances with different visual representations.

| Visu. Repr. | Avg. Succ. (↑) | Avg. Rank (↓) | Close Jar | Insert Peg | Install Bulb | Meat off Grill | Open Drawer | Place Cups | Sort Shape | Place Wine |
|---|---|---|---|---|---|---|---|---|---|---|
| LIV [26] | 61.0 ±0.4 | 3.6 ±1.7 | 68.0 ±17.3 | 4.0 ±3.3 | 41.3 ±21.7 | 96.0 ±5.7 | 100.0 ±0.0 | 1.3 ±1.9 | 32.0 ±5.7 | 96.0 ±3.3 |
| R3M [27] | 59.2 ±2.5 | 4.2 ±1.8 | 65.3 ±21.0 | 4.0 ±5.7 | 44.0 ±13.1 | 97.3 ±3.8 | 98.7 ±1.9 | 0.0 ±0.0 | 12.0 ±3.3 | 86.7 ±6.8 |
| VIP [35] | 56.5 ±2.0 | 4.0 ±2.0 | 72.0 ±14.2 | 2.7 ±1.9 | 38.7 ±15.4 | 93.3 ±9.4 | 100.0 ±0.0 | 5.3 ±5.0 | 22.7 ±10.0 | 89.3 ±8.2 |
| VC-1 [28] | 56.9 ±1.6 | 3.7 ±2.3 | 73.3 ±9.4 | 1.3 ±1.9 | 30.7 ±18.6 | 93.3 ±9.4 | 100.0 ±0.0 | 8.0 ±3.3 | 20.0 ±8.6 | 86.7 ±10.0 |
| CLIP [43] | 58.4 ±1.6 | 4.3 ±2.0 | 62.7 ±21.7 | 4.0 ±3.3 | 46.7 ±15.4 | 96.0 ±5.7 | 100.0 ±0.0 | 0.0 ±0.0 | 16.0 ±3.3 | 82.7 ±13.6 |
| DINOv2 [29] | 58.3 ±1.4 | 4.4 ±1.6 | 65.3 ±18.0 | 2.7 ±3.8 | 41.3 ±21.2 | 98.7 ±1.9 | 100.0 ±0.0 | 1.3 ±1.9 | 13.3 ±1.9 | 80.0 ±8.6 |
| ResNet [44] | 60.5 ±2.0 | 3.8 ±1.7 | 68.0 ±20.4 | 2.7 ±3.8 | 46.7 ±10.5 | 96.0 ±5.7 | 100.0 ±0.0 | 0.0 ±0.0 | 13.3 ±5.0 | 84.0 ±6.5 |

| Visu. Repr. | Push Buttons | Put in Cupboard | Put in Drawer | Put in Safe | Drag Stick | Slide Block | Stack Blocks | Stack Cups | Sweep to Dustpan | Turn Tap |
|---|---|---|---|---|---|---|---|---|---|---|
| LIV [26] | 78.7 ±8.2 | 57.3 ±3.8 | 97.3 ±1.9 | 97.3 ±3.8 | 88.0 ±8.6 | 73.3 ±3.8 | 4.0 ±3.3 | 1.3 ±1.9 | 66.7 ±5.0 | 94.7 ±5.0 |
| R3M [27] | 89.3 ±5.0 | 50.7 ±10.0 | 85.3 ±5.0 | 94.7 ±5.0 | 94.7 ±5.0 | 82.7 ±12.4 | 8.0 ±3.3 | 1.3 ±1.9 | 53.3 ±8.2 | 97.3 ±3.8 |
| VIP [35] | 92.0 ±3.3 | 64.0 ±8.6 | 93.3 ±3.8 | 89.3 ±10.0 | 10.7 ±7.5 | 46.7 ±36.7 | 2.7 ±1.9 | 5.3 ±3.8 | 92.0 ±8.6 | 97.3 ±1.9 |
| VC-1 [28] | 93.3 ±5.0 | 65.3 ±8.2 | 93.3 ±6.8 | 92.0 ±8.6 | 9.3 ±6.8 | 52.0 ±31.5 | 4.0 ±3.3 | 9.3 ±7.5 | 92.0 ±8.6 | 100.0 ±0.0 |
| CLIP [43] | 89.3 ±5.0 | 46.7 ±13.2 | 81.3 ±3.8 | 94.7 ±5.0 | 94.7 ±5.0 | 81.3 ±10.5 | 10.7 ±3.8 | 5.3 ±5.0 | 52.0 ±8.6 | 88.0 ±14.2 |
| DINOv2 [29] | 88.0 ±3.3 | 50.7 ±15.4 | 85.3 ±5.0 | 94.7 ±5.0 | 94.7 ±5.0 | 78.7 ±6.8 | 9.3 ±1.9 | 4.0 ±3.3 | 46.7 ±5.0 | 94.7 ±7.5 |
| ResNet [44] | 93.3 ±1.9 | 61.3 ±9.4 | 98.7 ±1.9 | 90.7 ±8.2 | 86.7 ±10.5 | 73.3 ±6.8 | 17.3 ±11.5 | 1.3 ±1.9 | 56.0 ±5.7 | 100.0 ±0.0 |

**Table 10:** Multi-task performance with different weighting parameter $\alpha$.

| $\alpha$ | Avg. Succ. (↑) | Avg. Rank (↓) | Close Jar | Insert Peg | Install Bulb | Meat off Grill | Open Drawer | Place Cups | Sort Shape | Place Wine |
|---|---|---|---|---|---|---|---|---|---|---|
| 0 | 57.3 ±2.1 | 2.8 ±1.0 | 74.7 ±10.0 | 0.0 ±0.0 | 32.0 ±18.2 | 52.0 ±14.2 | 98.7 ±1.9 | 6.7 ±5.0 | 29.3 ±5.0 | 81.3 ±5.0 |
| 0.5 | 57.6 ±2.2 | 2.7 ±0.8 | 73.3 ±10.5 | 0.0 ±0.0 | 33.3 ±11.5 | 49.3 ±21.0 | 100.0 ±0.0 | 5.3 ±3.8 | 29.3 ±6.8 | 92.0 ±5.7 |
| 1 | 61.0 ±0.4 | 2.3 ±1.4 | 68.0 ±17.3 | 4.0 ±3.3 | 41.3 ±21.7 | 96.0 ±5.7 | 100.0 ±0.0 | 1.3 ±1.9 | 32.0 ±5.7 | 96.0 ±3.3 |
| 2 | 58.0 ±2.3 | 2.2 ±0.8 | 76.0 ±9.8 | 0.0 ±0.0 | 33.3 ±10.5 | 48.0 ±11.3 | 100.0 ±0.0 | 8.0 ±3.3 | 29.3 ±3.8 | 88.0 ±6.5 |

| $\alpha$ | Push Buttons | Put in Cupboard | Put in Drawer | Put in Safe | Drag Stick | Slide Block | Stack Blocks | Stack Cups | Sweep to Dustpan | Turn Tap |
|---|---|---|---|---|---|---|---|---|---|---|
| 0 | 90.7 ±5.0 | 62.7 ±12.4 | 96.0 ±5.7 | 77.3 ±3.8 | 76.0 ±11.8 | 58.7 ±9.4 | 18.7 ±12.4 | 1.3 ±1.9 | 78.7 ±16.4 | 96.0 ±3.3 |
| 0.5 | 85.3 ±6.8 | 62.7 ±13.2 | 96.0 ±3.3 | 80.0 ±0.0 | 76.0 ±14.2 | 58.7 ±6.8 | 17.3 ±13.6 | 0.0 ±0.0 | 80.0 ±15.0 | 97.3 ±1.9 |
| 1 | 78.7 ±8.2 | 57.3 ±3.8 | 97.3 ±1.9 | 97.3 ±3.8 | 88.0 ±8.6 | 73.3 ±3.8 | 4.0 ±3.3 | 1.3 ±1.9 | 66.7 ±5.0 | 94.7 ±5.0 |
| 2 | 88.0 ±8.6 | 60.0 ±9.8 | 100.0 ±0.0 | 81.3 ±6.8 | 77.3 ±12.4 | 58.7 ±6.8 | 14.7 ±10.0 | 1.3 ±1.9 | 84.0 ±17.3 | 96.0 ±5.7 |

decomposed into 10 sub-tasks, the mainstream policy training datasets typically have 10 million sub-tasks. ($\approx$ 10 million entries in the database). **The Open X-Embodiment (OXE) Dataset [47]:** A landmark collaboration among 21 institutions, OXE provides over 1 million robot trajectories from 22 different robot embodiments. Its explicit goal is to foster the development of generalist models, demonstrating that the community is actively removing the proprietary data barriers of the past. The explicit purpose of OXE is to provide a standardized, large-scale resource to train generalist models that have demonstrated significant performance gains by training on this diverse data. **Hugging Face**

**Table 11:** Multi-task performance with different numbers of demonstrations.

| Demo. Num. | Avg. Succ. (↑) | Avg. Rank (↓) | Close Jar | Insert Peg | Install Bulb | Meat off Grill | Open Drawer | Place Cups | Sort Shape | Place Wine |
|---|---|---|---|---|---|---|---|---|---|---|
| 1 (RDD) | 59.1 ±3.4 | 1.8 ±0.8 | 75.6 ±11.1 | 6.2 ±5.7 | 35.6 ±9.5 | 65.8 ±20.9 | 100.0 ±0.0 | 5.8 ±4.7 | 25.8 ±3.8 | 91.1 ±7.7 |
| 3 (RDD) | 61.0 ±0.4 | 1.8 ±0.7 | 68.0 ±17.3 | 4.0 ±3.3 | 41.3 ±21.7 | 96.0 ±5.7 | 100.0 ±0.0 | 1.3 ±1.9 | 32.0 ±5.7 | 96.0 ±3.3 |
| 3 (UVD [25]) | 57.1 ±0.3 | 2.3 ±0.6 | 66.7 ±13.2 | 4.0 ±5.7 | 37.3 ±19.1 | 93.3 ±9.4 | 100.0 ±0.0 | 2.7 ±1.9 | 21.3 ±10.5 | 77.3 ±11.5 |

| Demo. Num. | Push Buttons | Put in Cupboard | Put in Drawer | Put in Safe | Drag Stick | Slide Block | Stack Blocks | Stack Cups | Sweep to Dustpan | Turn Tap |
|---|---|---|---|---|---|---|---|---|---|---|
| 1 (RDD) | 86.7 ±5.7 | 60.4 ±11.8 | 97.8 ±2.0 | 78.2 ±13.3 | 61.8 ±28.7 | 79.6 ±16.2 | 11.6 ±8.1 | 2.2 ±2.7 | 87.1 ±16.2 | 92.9 ±14.7 |
| 3 (RDD) | 78.7 ±8.2 | 57.3 ±3.8 | 97.3 ±1.9 | 97.3 ±3.8 | 88.0 ±8.6 | 73.3 ±3.8 | 4.0 ±3.3 | 1.3 ±1.9 | 66.7 ±5.0 | 94.7 ±5.0 |
| 3 (UVD [25]) | 62.7 ±12.4 | 44.0 ±6.5 | 84.0 ±6.5 | 96.0 ±5.7 | 85.3 ±13.2 | 82.7 ±12.4 | 16.0 ±3.3 | 1.3 ±1.9 | 60.0 ±16.3 | 93.3 ±5.0 |

**Table 12:** Multi-task performance of Vanilla Planner without finetuning on the target task.

| Method | Avg. Succ. (↑) | Avg. Rank (↓) | Meat off Grill | Open Drawer | Place Wine | Push Buttons | Put in Cupboard |
|---|---|---|---|---|---|---|---|
| w/o finetuning on target task | 77.9 ±4.3 | 1.6 ±0.5 | 99.2 ±2.4 | 99.6 ±1.2 | 86.4 ±8.8 | 70.4 ±8.0 | 61.2 ±16.8 |
| RDD (Ours) | 79.6 ±7.2 | 1.4 ±0.5 | 84.4 ±21.1 | 99.2 ±2.4 | 86.4 ±15.4 | 84.0 ±7.8 | 41.2 ±17.1 |

| Method | Put in Drawer | Put in Safe | Drag Stick | Slide Block | Sweep to Dustpan | Turn Tap |
|---|---|---|---|---|---|---|
| w/o finetuning on target task | 86.0 ±14.3 | 94.8 ±9.0 | 74.0 ±23.3 | 62.4 ±16.8 | 30.0 ±15.3 | 92.4 ±14.4 |
| RDD (Ours) | 97.2 ±3.1 | 98.4 ±3.2 | 68.0 ±25.0 | 65.2 ±14.3 | 57.2 ±29.7 | 94.0 ±5.1 |

**Table 13:** Comparing RDD with Gemini-2.5-pro.

| Method | Avg. Succ. (↑) | Avg. Rank (↓) | Close Jar | Install Bulb | Meat off Grill | Open Drawer | Place Wine | Push Buttons |
|---|---|---|---|---|---|---|---|---|
| Gemini-2.5-pro | 72.6 ±4.7 | 1.7 ±0.4 | 41.2 ±30.1 | 40.8 ±16.5 | 83.2 ±15.2 | 99.6 ±1.2 | 86.4 ±11.1 | 82.4 ±8.6 |
| RDD (Ours) | 74.9 ±6.9 | 1.3 ±0.4 | 46.0 ±28.2 | 52.8 ±16.4 | 84.4 ±21.1 | 99.2 ±2.4 | 86.4 ±15.4 | 84.0 ±7.8 |

| Method | Put in Cupboard | Put in Drawer | Put in Safe | Drag Stick | Slide Block | Sweep to Dustpan | Turn Tap |
|---|---|---|---|---|---|---|---|
| Gemini-2.5-pro | 38.4 ±10.6 | 94.0 ±6.8 | 93.6 ±9.2 | 73.6 ±22.3 | 63.6 ±14.4 | 48.4 ±14.9 | 99.2 ±2.4 |
| RDD (Ours) | 41.2 ±17.1 | 97.2 ±3.1 | 98.4 ±3.2 | 68.0 ±25.0 | 65.2 ±14.3 | 57.2 ±29.7 | 94.0 ±5.1 |

**Table 14:** Performance of FAISS nearest neighbor search and RDD time on NVIDIA 4090.

| Hardware | Dim | Vec Num | QPS | $L_{max}$ | $L_{\mathcal{I}}$ | RDD Time (s) |
|---|---|---|---|---|---|---|
| NVIDIA 4090 | 2048 | 10M | 386 | 100 | 500 | 115 (4.3 fps) |

**SmolVLA Dataset [48]:** The emergence of models like SmolVLA, a capable vision-language-action model trained entirely on 23k episodes from 487 open-sourced community datasets through the LeRobot framework, outperforms the closed-source-dataset policy $\pi_0$ [4]. **AgiBot World [33]:** AgiBot World provides not just datasets but complete open-source toolchains and standardized data collection pipelines, further enriching the public ecosystem. It has collected over 1 million trajectories on over 100 homogeneous robots. Their proposed model GO-1, entirely trained on this open-sourced dataset, outperforms the closed-source dataset policy $\pi_0$ [4].

# E   Broader Impacts

The potential negative societal impacts of our work are consistent with those commonly observed in robotics research, including risks related to privacy, labor displacement, and unintended misuse in sensitive contexts. While our method is primarily designed to enhance the scalability and efficiency of robotic systems, such advancements may accelerate deployment in real-world settings, amplifying both positive and negative consequences. In parallel, advances in point cloud analysis [49, 50],

cooperative motion prediction [51], autonomous driving frameworks [52, 53, 54, 55], and generative AI for driving [56] highlight both the promise and the risks of deploying increasingly capable vision-action models. Broader surveys of visual foundation models [57] and new work on multimodal alignment [58, 59] further strengthen the importance of trustworthy design and governance, especially for safety-critical applications such as transportation and human-robot interaction [60, 61]. To mitigate these risks, we emphasize alignment with ethical guidelines, including fairness, accountability, transparency, and safety, and encourage interdisciplinary collaboration to monitor societal impacts as these technologies evolve [62].

