# OpenReview forum: "RDD: Retrieval-Based Demonstration Decomposer for Planner Alignment in Long-Horizon Tasks"
_NeurIPS.cc/2025/Conference — NeurIPS 2025 poster_

### Official Review · Reviewer_G7RJ · 2025-06-23

**Clarity:** 4
**Significance:** 3
**Originality:** 4
**Rating:** 4
**Confidence:** 3

**Summary:**

This work introduced RDD, which can decompose demonstrations into sub-tasks without relying on human annotation or heuristics rules. RDD aligns the visual features of sub-task intervals with the training data of visuomotor policy. The author provides detailed demo video and code, which significantly enhances the reproducibility of the proposed method. RDD achieves good performance on RLBench and robustness.

**Questions:**

See Strengths And Weaknesses

**Ethical Concerns:**

["NO or VERY MINOR ethics concerns only"]

**Final Justification:**

The authors’ responses have addressed my concerns. I will maintain my scores, i.e, borderline accept.

**Limitations:**

Yes

**Quality:**

3

**Strengths And Weaknesses:**

Strength:
1. The author provides detailed demo video and code, which significantly enhances the reproducibility of the proposed method.
2. The author conduct ablations and present detailed analysis for the proposed method.
3. The paper is well-written.

Weaknesses:
1. How does the RDD perform compared to the decomposer based on code-generating LLM/VLMs, e.g., instruct2act [1], robotics programmer [2] and Code as policies [3].

[1] Huang, Siyuan, et al. "Instruct2act: Mapping multi-modality instructions to robotic actions with large language model." arXiv preprint arXiv:2305.11176 (2023).

[2] Xie, Senwei, et al. "Robotic Programmer: Video Instructed Policy Code Generation for Robotic Manipulation." arXiv preprint arXiv:2501.04268 (2025).

[3] Liang, Jacky, et al. "Code as policies: Language model programs for embodied control." ICRA, 2023.

---

> ### Author Rebuttal · Authors · 2025-07-30
>
> First of all, we sincerely thank the reviewer for the insightful feedback. We are encouraged that the reviewer pointed out the major strengths of our paper: 1) the paper’s clarity; 2) detailed analysis; and 3) reproducibility. We address the reviewer’s comments and questions point by point in the following.
>
> # Response to Weakness 1
>
> Thanks for providing the reference papers about VLM-based sub-task decomposition. **We would like to clarify that these works use code scripts as low-level control policies, which is different from our settings, where a learning-based visuomotor policy is the target for planner alignment.** Instead, we provide extended experiments by using the state-of-the-art large multi-model language model Gemini-2.5-pro as a sub-task decomposer for comparison. In the following table, we compared RDD with a Gemini-2.5-pro-based decomposer with the following prompt:
>
> > There is a robot doing a task, which can be segmented into multiple steps.
> > A keyframe is where the robot finishes the previous step and begins the next.
> > Can you help me find ALL indexes of keyframes?
> > Please return a list of indices, for example: [15, 65, 105, ...].
> > Note that the frame index starts from 0 instead of 1.
>
> | Method         | Avg. Success Rate (↑) | Avg. Ranking (↓) |
> | -------------- | --------------------- | ---------------- |
> | Gemini-2.5-pro | 72.6 ± 4.7            | 1.7 ± 0.4        |
> | RDD (Ours)     | **74.9 ± 6.9**        | **1.3 ± 0.4**    |
>
> As shown in the table, RDD outperforms Gemini-2.5-pro despite its powerful general video understanding abilities and significant computing resources. **These results highlight the necessity of planner alignment, as well as the effectiveness of our approach RDD.**
>
> We will also include and discuss the reference papers provided by the reviewer in the related work section of the revised paper.

---

> ### Comment · Reviewer_G7RJ · 2025-08-04
>
> The authors' response has addressed my concerns. I will maintain my score, i.e., borderline accept for the paper.

---

> > ### Author Response · Authors · 2025-08-04
> > **Thank You for Championing Acceptance**
> >
> > Dear Reviewer G7RJ,
> >
> > Thank you for your support and for championing the acceptance of our work.
> >
> > Best regards,
> > Authors

---

### Official Review · Reviewer_wuNw · 2025-07-03

**Clarity:** 3
**Significance:** 2
**Originality:** 3
**Rating:** 4
**Confidence:** 4

**Summary:**

This paper proposes a method to decompose the demonstration into "sub-tasks" to improve the learning of VLA models. The authors formulate the decomposition as an optimal partition problem, where the partition objective relates to the learning data of low-level policies. The decomposition method is applied to a baseline hierarchical VLA and tested on 18 RLBench tasks for evaluation.

**Questions:**

See the strength and weakness section.

**Ethical Concerns:**

["NO or VERY MINOR ethics concerns only"]

**Final Justification:**

The author rebuttal has addressed most of my concerns. Therefore, I'm increasing my score to an accept.

**Limitations:**

yes

**Paper Formatting Concerns:**

No formatting concerns.

**Quality:**

2

**Strengths And Weaknesses:**

[+] The overall idea is well-motivated and shares a lot of insights with recent hierarchical VLA models, where we can use long-horizon task demonstrations to facilitate the learning of plan and action, and also perhaps task augmentation.

[+] The overall description of the framework is clear and illustrative, with major points of the design easy and clear to follow.

[-] One major concern lies in the marginal improvement of this method to the tasks selected in the evaluation phase. The current result is a bit hard to validate the effectiveness of the current method, especially given several ties between the proposed and the Uniform or Heuristic methods.

[-] Another point is that I do feel the current framework can be extended to form many more interesting attempts and analyses, e.g., cross-domain or sim2real transfer (as the current decomposition should ideally support skill reuse). The current discussion on only RLBench is a bit narrow in my opinion, especially given the marginal improvement.

---

> ### Author Rebuttal · Authors · 2025-07-30
>
> First of all, we sincerely thank the reviewer for the insightful feedback. We are encouraged that the reviewer pointed out the strengths of our paper:  the strong motivation for our idea and the clarity of our paper. We address the reviewer’s comments and questions point by point in the following.
>
> # Response to Weakness 1
>
> We sincerely thank the reviewer for the comment. We respectfully clarify that our work's significance is threefold:
>
> 1. **Near-oracle Performance and State-of-the-art Advancement:** We would like to emphasize that the Heuristic baseline, which relies on manually-defined rules, represents a strong, near-oracle method for these tasks. Our proposed RDD, without any task-specific knowledge, **achieves comparable performance to this oracle**, as shown in Table 1. This is a significant result, demonstrating the effectiveness of our approach. Furthermore, RDD improves the average task success rate by 6.7% over the state-of-the-art general-purpose decomposer, UVD. **To better demonstrate the effectiveness of RDD, we provide extended experiment results at the end of this rebuttal.**
>
> 2. **Scalability and Growing Importance:** More importantly, we argue that the true advantage of RDD also lies in its scalability and adaptability. The current trend in robotics is toward leveraging large-scale, diverse visuomotor datasets. In such scenarios, designing effective, task-specific heuristics will become increasingly complex and impractical. RDD is designed to automatically and efficiently align the planner with the visuomotor policy’s training data, **a feature that will become crucial as the complexity of tasks and datasets grows.** In section 6 of Hi Robot [A], the state-of-the-art hierarchical vision-language-action research, especially highlights this problem:
>
>    > "The training process decouples the high-level and low-level models, and they are not aware of one another’s capabilities except through the training examples. Coupling these two layers more directly, e.g., by allowing the high-level policy to be more aware of how successfully the low-level policy completes each command, would be an exciting direction for future work."
>
>    RDD provides an automated and efficient solution to align demonstrations with a policy's training data, which is a crucial capability for future complex hierarchical VLAs.
>
> 3. **Novel Problem Formulation and Efficient Solution:** As recognized and highlighted by Reviewer 9Wsv, **we are the first to identify and formally address the crucial alignment gap between high-level planner training data and the capabilities of the low-level visuomotor policy in hierarchical VLAs.** As we note in our paper, sub-tasks generated by heuristics can deviate significantly from what the visuomotor policy is optimized for, leading to degraded performance. Our work is the first to tackle this specific misalignment problem. **We introduce a novel formulation of this problem as an optimal partitioning task.** This provides a principled mathematical framework for demonstration decomposition. Based on this, we develop an efficient dynamic programming solver and analyze its complexity. We prove that **when the maximum interval length is bounded, our algorithm achieves linear time complexity O(N), which is a vital property for general robotics applications**. This theoretical analysis of the algorithm’s efficiency is a key algorithmic contribution.
>
> # Response to Weakness 2
>
> We thank the reviewer for this insightful question regarding RDD’s applications in broader scenarios. Both cross-domain and sim2real transfer require RDD to handle the out-of-distribution (OOD) cases, where fundamentally novel sub-tasks appear in the demonstrations to be decomposed. **In such OOD scenarios, the objective changes to: aligning sub-task intervals to both existing visuomotor sub-tasks and potentially new sub-tasks.** In this response, we show how RDD can extend to identify cross-domain novel sub-tasks and its performance in the real-world dataset.
>
> Thanks to the flexible design of RDD's scoring function $\tilde{J}$, we do have a simple yet effective mechanism to gracefully handle these OOD scenarios and prevent catastrophic failures. Our approach involves two stages: OOD identification and OOD-aware decomposition.
>
> **OOD Identification**
>
> First, we must determine if a given demonstration is, in fact, OOD. We can quantify the novelty of a demonstration by calculating $\\Delta$ , the average similarity score of the optimal partition $\\mathrm{P}$ found by the standard RDD algorithm:
>
>
> $$
> \\Delta=\\frac{1}{|\\mathrm{P}|}\\sum_{\\mathcal{I} \\in \\mathrm{P}} \\tilde{J}(\\mathcal{I}).
> $$
>
> A low value of *$\\Delta$* indicates that even the best possible decomposition has a low similarity to the sub-tasks in the training data, which signals an OOD demonstration. A simple threshold can be used to trigger our adaptive mechanism.
>
> **OOD-aware Decomposition**
>
> Once an OOD demonstration is identified, we introduce a modified similarity measure, $\\mathbf{sim}_{OOD}$, which incorporates a general interval scoring function $\\mathbf{G}$:
>
>
> $$
> \\mathbf{sim}_{OOD}(\\mathcal{I}_j^i,\\tilde{\\mathcal{I}_j^i})=\\mathbf{sim}(\\mathcal{I}_j^i,\\tilde{\\mathcal{I}_j^i})+\\beta \\mathbf{G}(\\mathcal{I}_j^i).
> $$
>
> Here, $\\mathbf{G}$ evaluates how well a proposed interval aligns with "general" sub-tasks, independent of the planner's specific training set. The hyperparameter $\\beta$ balances the trade-off between aligning with known sub-tasks (the original RDD objective) and discovering novel, generalizable sub-tasks. $\\mathbf{G}$ can be implemented using functions like UVD [B] to measure how well an interval conforms to generic change-point detection heuristics:
>
> $$
> \\mathbf{G}(\\mathcal{I})=-\\frac{1}{|\\mathcal{I}|}\\mathbf{abs}(b-\\mathbf{UVD}(e,\\mathcal{I})),
> $$
>
> where $b,e$ represent the index of beginning and ending frame of interval $\\mathcal{I}$. $\\mathbf{UVD}(e,\\mathcal{I})$ gives the index of the UVD predicted beginning frame, given the goal frame on $e$. This $\\mathbf{G}$ essentially measures how $\\mathcal{I}$ aligns with the general sub-task interval proposed by UVD. This function penalizes intervals that deviate from what a general-purpose decomposition model would propose, effectively acting as a robust prior. The newly discovered sub-tasks can then be used to fine-tune both the planner and the visuomotor policy, which allows the system to learn and adapt to the new skills.
>
> ## Additional Experimental Validation on LIBERO
>
> To validate this approach, we conducted additional experiments on a challenging LIBERO benchmark. To simulate a highly OOD scenario, we used the human-operated demonstration dataset from RoboCerebra [C], where demos are highly diverse in objects, task goals, and arrangements. We evaluated the quality of the decomposition against ground-truth segmentations using the mean Intersection-over-Union (mIoU) (see response to Reviewer DAU1 for details). We use 560 demos to build the RDD database and test on the remaining 140 demos.
>
> | Methods             | mIoU (LIBERO OOD Test) |
> | ------------------- | --------------- |
> | UVD                 | 0.598           |
> | RDD ($\\beta=0.25$) | 0.624 ( ↑ 4.4%) |
> | RDD ($\\beta=0.1$)  | 0.630 ( ↑ 5.3%) |
> | RDD ($\\beta=0.05$) | 0.614 ( ↑ 2.6%) |
>
> The results clearly show that our OOD-aware **RDD not only avoids catastrophic failure but outperforms the baseline UVD decomposer in OOD settings.** By balancing retrieval with a general sub-task prior, RDD can successfully leverage even limited similarities in novel scenarios to produce a superior decomposition. This demonstrates the robustness and adaptability of the RDD framework.
>
> ## Additional Experimental Validation on Real-World Dataset
>
> For in-distribution aligning (original goal of RDD), we test RDD on the real-world manipulation dataset AgiBotWorld-Alpha [D] to show the generalization ability of RDD visual features.  We test RDD and UVD on the task "supermarket" where 152 demos are used to build RDD database and 37 demos are used for testing.
>
> | Methods | mIoU (AgiBotWorld Alignment Test) |
> | ------- | --------------------------- |
> | UVD     | 0.506                       |
> | **RDD**     | **0.706 ( ↑ 39.6%)**            |
>
> RDD significantly outperforms the baseline method UVD, proving its generalization ability.
>
> We will add the above discussions to the revised paper.
>
> # References
>
> [A] Shi, et al. "Hi robot: Open-ended instruction following with hierarchical vision-language-action models." arXiv preprint (2025).
>
> [B] Zhang, et al. "Universal visual decomposer." ICRA, 2024.
>
> [C] Han, et al. "RoboCerebra." arXiv preprint  (2025).
>
> [D] Bu, et al. "Agibot world colosseo." arXiv preprint (2025).

---

> ### Author Response · Authors · 2025-08-06
> **Follow-Up: Confirming We Have Fully Addressed Your Comments**
>
> Dear Reviewer wuNw,
>
> I hope this message finds you well. As the discussion window draws to a close, we would like to ensure that we have fully addressed your concerns. If you have any further feedback or questions, please let us know while the window remains open so we can respond before the deadline. Your insights are invaluable to improving our work. Thank you very much for your time and thoughtful review.
>
> Sincerely,
> Authors

---

### Official Review · Reviewer_DAU1 · 2025-07-03

**Clarity:** 3
**Significance:** 2
**Originality:** 2
**Rating:** 4
**Confidence:** 4

**Summary:**

This paper identifies a critical alignment issue between high-level planners and low-level visuomotor policies in hierarchical vision-language-action (VLA) agents. To address this, the authors propose a Retrieval-based Demonstration Decomposer (RDD), a novel training-free method to automatically segment long-horizon demonstrations for planner fine-tuning. The core mechanism of RDD is to formulate the task as an optimal partitioning problem, which is solved by retrieving visually and temporally similar sub-tasks from the low-level policy's training data to guide the decomposition.

**Questions:**

1. Given RDD's reliance on the D_aug_train database, how does its performance depend on the quality and style of the upstream heuristic used to create that database? Is RDD robust to different or lower-quality source heuristics?
2. What is the expected behavior of RDD when encountering a truly novel sub-task not present in its retrieval database? Have you considered fallback mechanisms for such low-confidence retrieval scenarios?
3. To better isolate the contribution, have you considered a more direct evaluation of decomposition quality itself？

**Ethical Concerns:**

["NO or VERY MINOR ethics concerns only"]

**Final Justification:**

The authors' response has addressed some of my concerns, and I encourage the authors to include these experiments and discussions in the paper. I will raise my score to borderline accept.

**Limitations:**

Yes, the authors have acknowledged the existence of a limitations section in the appendix. However, given the significance of certain limitations (e.g., the reliance on the quality of the policy's training data), it would improve the paper to briefly discuss the most critical ones in the main text for better transparency.

**Quality:**

2

**Strengths And Weaknesses:**

Strengths
1. The paper tackles a highly relevant and significant problem in hierarchical robot learning. Ensuring that the high-level planner generates instructions that the low-level policy can reliably execute is crucial for the success of long-horizon tasks. The proposed direction of aligning the planner's fine-tuning data with the policy's existing capabilities is a valuable contribution to the field.
2. The central concept of RDD is elegant and highly intuitive. Instead of relying on hand-engineered heuristics or expensive human annotations, RDD leverages the policy's own training data as a "template" for what constitutes a good, executable sub-task.
3. The experimental evaluation is comprehensive, featuring strong baselines on a standard benchmark and extensive ablations that validate key design choices and demonstrate the method's effectiveness.

Weakness
1. The sub-task representation, using only start and end frames, is simplistic. It discards crucial intermediate dynamics and may fail to distinguish between sub-tasks with different motions but similar boundary states.
2. The technical novelty is somewhat limited. The method is primarily a novel application and combination of existing techniques (dynamic programming, visual similarity search) rather than a fundamental algorithmic contribution.
3. The retrieval-based nature of RDD raises questions about its robustness to novel, out-of-distribution sub-tasks. The paper lacks discussion and experiments for scenarios where a new demonstration contains actions not well-represented in the training database.

---

> ### Author Rebuttal · Authors · 2025-07-30
>
> First of all, we sincerely thank the reviewer for the insightful feedback. We are encouraged that the reviewer pointed out the strengths of our paper: 1) the significance of the problem we address; 2) the elegant and intuitive nature of our proposed method; and 3) the comprehensive experimental validation with extensive ablations. We address the reviewer’s comments and questions point by point in the following.
>
> # Response to Weakness 1
>
> We thank the reviewer for this insightful question. A key strength of our RDD framework is its modularity. The interval similarity measure, **sim**, is a plug-and-play component that offers a convenient and flexible interface to incorporate domain-specific knowledge or more powerful representations without altering the core dynamic programming algorithm. For instance, one could easily implement a more sophisticated **sim** function using powerful video encoders like VideoCLIP [A], to capture richer temporal and semantic information. Our novel, general problem formulation, along with the flexible design of our scoring function, is ready to integrate various designs of similarity measures.
>
> In "Response to Question 2 & Weakness 3" we provide experimental validation of an implementation of this idea.
>
> # Response to Weakness 2
>
> We thank the reviewer for their valuable feedback. We respectfully emphasize that our work's primary contribution is the novel formulation and solution to the critical, previously unaddressed alignment gap between a hierarchical VLA's high-level planner and its low-level visuomotor policy. Our novelty is threefold:
>
> 1. **Novel Problem Formulation:** As noted by Reviewer 9Wsv, we are the first to formally identify and address the misalignment between a planner's generated sub-tasks and the visuomotor policy's capabilities. We introduce a principled mathematical framework by formulating this as an optimal partitioning problem.
> 2. **A Provable and Linear Complexity Algorithm:** Crucially for robotics applications, we prove that it is possible for the dynamic programming algorithm to achieve linear time complexity, **O(N)**, when the maximum interval length is bounded. (Corollary3.1.1.)
> 3. **Unbiased Scoring Function:** We propose a novel interval scoring function to measure the semantic alignment of sub-tasks with the policy's training data. Its additive property not only enables the dynamic programming solution but, as proven in Proposition 3.1, it also prevents segmentation bias, ensuring the quality of the partitioning.
>
> We believe the identification of this fundamental problem, novel mathematical formulation and an efficient, theoretically-grounded algorithm, constitutes a significant contribution. We will highlight these points more clearly in the revised paper.
>
> # Response to Question 1
>
> We thank the reviewer for their insightful question regarding RDD’s sensitivity to the quality of sub-task segmentation. We would like to address this by highlighting two key aspects of our approach:
>
> First, we wish to clarify RDD’s primary objective. The core contribution is not to learn new skills or correct flawed ones, but to enable a high-level planner to effectively utilize the skills that the low-level visuomotor policy already possesses. Therefore, **RDD is largely agnostic to the "optimality" of the skills themselves. This ensures the planner generates commands that the policy can reliably execute, rather than potentially "better" ones it cannot handle.**
>
> Second, in scenarios where the visuomotor policy's training data contains significant noisy samples that the policy fails to learn, **RDD can be easily integrated with dataset curation techniques [B, C].** These methods can serve as a pre-processing step to filter the visuomotor training set. For instance, CUPID [B] computes an "action influence" score for state-action pairs that can be used to evaluate each segment's contribution to the policy's final behavior. By applying a simple threshold, low-influence or flawed segments can be pruned from the dataset before RDD uses it as a reference. This would prevent catastrophic failures by ensuring RDD aligns demonstrations only with high-quality, influential sub-tasks.
>
> We will add these clarifying discussions to the revised paper.
>
> # Response to Question 2 & Weakness 3
>
> We thank the reviewer for this insightful question regarding RDD’s handling of out-of-distribution (OOD) demonstrations. While the primary objective of RDD is to align the planner with the visuomotor policy’s existing capabilities, this is a critical aspect for real-world applicability. **In such OOD scenarios, the objective changes to: aligning sub-task intervals to both existing visuomotor sub-tasks and potentially new sub-tasks.**
>
> Thanks to the flexible design of RDD's scoring function $\\tilde{J}$, we do have a simple yet effective mechanism to gracefully handle these OOD scenarios and prevent catastrophic failures. Our approach involves two stages: OOD identification and OOD-aware decomposition.
>
> **OOD Identification**
>
> First, we must determine if a given demonstration is, in fact, OOD. We can quantify the novelty of a demonstration by calculating $\\Delta$, the average similarity score of the optimal partition $\\mathrm{P}$ found by the standard RDD algorithm:
>
>
> $$
> \\Delta=\\frac{1}{|\\mathrm{P}|}\\sum_{\\mathcal{I} \\in \\mathrm{P}} \\tilde{J}(\\mathcal{I}).
> $$
>
> A low value of *$\\Delta$* indicates that even the best possible decomposition has a low similarity to the sub-tasks in the training data, which signals an OOD demonstration. A simple threshold can be used to trigger our adaptive mechanism.
>
> **OOD-aware Decomposition**
>
> Once an OOD demonstration is identified, we introduce a modified similarity measure, $\\mathbf{sim}_{OOD}$, which incorporates a general interval scoring function $\\mathbf{G}$:
>
>
> $$
> \\mathbf{sim}_{OOD}(\\mathcal{I}_j^i,\\tilde{\\mathcal{I}_j^i})=\\mathbf{sim}(\\mathcal{I}_j^i,\\tilde{\\mathcal{I}_j^i})+\\beta \\mathbf{G}(\\mathcal{I}_j^i).
> $$
>
> Here, $\\mathbf{G}$ evaluates how well a proposed interval aligns with "general" sub-tasks, independent of the planner's specific training set. The hyperparameter $\\beta$ balances the trade-off between aligning with known sub-tasks (the original RDD objective) and discovering novel, generalizable sub-tasks. $\\mathbf{G}$ can be implemented using functions like UVD [D] to measure how well an interval conforms to generic change-point detection heuristics:
>
> $$
> \\mathbf{G}(\\mathcal{I})=-\\frac{1}{|\\mathcal{I}|}\\mathbf{abs}(b-\\mathbf{UVD}(e,\\mathcal{I})),
> $$
>
> where $b,e$ represent the index of the beginning and ending frame of interval $\\mathcal{I}$. $\\mathbf{UVD}(e,\\mathcal{I})$ gives the index of the UVD predicted beginning frame, given the goal frame on $e$.
>
> ## Additional Experimental Validation on LIBERO
>
> For OOD scenarios, we conducted additional experiments on LIBERO. To simulate a highly OOD scenario, we used the human-operated demonstration dataset from RoboCerebra [E] where demos are highly diverse in objects, task goals, and arrangements. We evaluated the quality of the decomposition against ground-truth segmentations using the mean Intersection-over-Union (mIoU) (see response to Reviewer DAU1 for details). We use 560 demos to build the RDD database and test on the remaining 140 demos.
>
> | Methods             | mIoU (LIBERO OOD Test) |
> | ------------------- | ----------------- |
> | UVD                 | 0.598             |
> | RDD ($\\beta=0.25$) | 0.624 ( ↑ 4.4%)   |
> | RDD ($\\beta=0.1$)  | 0.630 ( ↑ 5.3%)   |
> | RDD ($\\beta=0.05$) | 0.614 ( ↑ 2.6%)   |
>
> The results clearly show that our OOD-aware **RDD not only avoids catastrophic failure but also outperforms the baseline UVD decomposer in OOD settings.** RDD can successfully leverage even limited similarities in novel scenarios to produce a superior decomposition.
>
> ## Additional Experimental Validation on Real-World Dataset
>
> For in-distribution alignment (original goal of RDD), we test RDD on the real-world manipulation dataset AgiBotWorld-Alpha [F] to show the generalization ability of RDD visual features.  We test RDD and UVD on the task "supermarket" where 152 demos are used to build the RDD database and 37 demos are used for testing.
>
> | Methods | mIoU (AgiBotWorld Alignment Test) |
> | ------- | --------------------------- |
> | UVD     | 0.506                       |
> | RDD     | 0.706 ( ↑ 39.6%)            |
>
> RDD significantly outperforms the baseline method UVD, proving its generalization ability.
>
> We will add the above discussions to the revised paper.
>
> # Response to Question 3
>
> We sincerely thank the reviewer for this insightful suggestion. To achieve this, we adopt the mean Intersection-over-Union (mIoU) as the metric to evaluate decomposition accuracy. We treat the segments generated by the task-specific decomposer, which was used to create the visuomotor policy’s training set, as the ground truth. The accuracy is formulated as
> $$
> accuracy=\\frac{1}{|\\mathrm{P}|}\\sum_{\\mathcal{I}\\in\\mathrm{P}} \\mathbf{IoU}(\\hat{\\mathcal{I}},\\tilde{\\mathcal{I}}),
> $$
> where $\\tilde{\\mathcal{I}}$ is the ground truth interval with maximum IoU with the predicted one $\\hat{\\mathcal{I}}$. To validate this metric, the table below shows the measured mIoU and end-to-end success rate of UVD and RDD.
>
> | Methods | mIoU    | Success Rates |
> | ------- | ------- | ------------- |
> | UVD     | 0.522   | 57.1          |
> | RDD     | 0.623 ↑ | 61.0 ↑        |
>
>
> # References
>
> [A] Xu, et al. "VideoCLIP" arXiv preprint (2021).
>
> [B] Agia, et al. "CUPID: Curating Data your Robot Loves with Influence Functions." *RSS.* 2025.
>
> [C] Hejna, et al. "Robot data curation with mutual information estimators." arXiv preprint (2025).
>
> [D] Zhang, et al. "Universal visual decomposer" ICRA, 2024.
>
> [E] Han, et al. "RoboCerebra" arXiv preprint  (2025).
>
> [F] Bu, et al. "Agibot world colosseo." arXiv preprint (2025).

---

> > ### Comment · Reviewer_DAU1 · 2025-08-04
> >
> > The authors' response has addressed some of my concerns, and I encourage the authors to include these experiments and discussions in the paper. I will raise my score to borderline accept.

---

### Official Review · Reviewer_9Wsv · 2025-07-03

**Clarity:** 3
**Significance:** 3
**Originality:** 3
**Rating:** 4
**Confidence:** 3

**Summary:**

This paper introduces RDD (Retrieval-based Demonstration Decomposer), a training-free method for decomposing long-horizon robotic demonstrations into sub-tasks for finetuning hierarchical Vision-Language-Action (VLA) models. The key insight is that existing decomposition methods (human annotation or heuristics) often generate sub-tasks that deviate from the training data of low-level visuomotor policies, leading to suboptimal performance. RDD addresses this by formulating demonstration decomposition as an optimal partitioning problem that explicitly aligns decomposed sub-tasks with the visuomotor policy's training set through visual feature retrieval. The method uses dynamic programming to efficiently solve the optimization problem and demonstrates 6.7% improvement over state-of-the-art methods on 18 RLBench tasks.

**Questions:**

1. How does RDD handle demonstrations containing sub-tasks that are fundamentally novel and have no close visual or temporal analogues in the policy's training set? Could you discuss this potential failure case and how it might impact the fine-tuning of the planner and overall task success? A change in my score would depend on a convincing explanation of how the system avoids catastrophic failure in such out-of-distribution scenarios.
2. Your method's success relies on the quality of the pre-existing visuomotor training set. How sensitive is RDD to the quality of the sub-task segmentation within this dataset? For example, if $\mathcal{D}^{train}_{aug}$ was segmented using a poor or noisy heuristic, would RDD simply learn to replicate this flawed strategy, or does the optimization process offer some resilience?
3. Can you provide examples where RDD fails to decompose demonstrations correctly? Specifically, are there cases where visually similar intervals have very different semantic meanings that confuse retrieval?

**Ethical Concerns:**

["NO or VERY MINOR ethics concerns only"]

**Final Justification:**

The rebuttal mostly solved my concerns that RDD can be extended to OOD scenarios and prevent catastrophic failures. I therefore maintain my initial rating of borderline accept.

**Limitations:**

Yes.

**Paper Formatting Concerns:**

No formatting issues.

**Quality:**

3

**Strengths And Weaknesses:**

=== Strengths ===

1. The paper identifies an important gap in hierarchical VLAs - the misalignment between planner training data and visuomotor policy capabilities. This is a practically relevant insight that is overlooked in prior work.
2. The formulation as an optimal partitioning problem with interval-wise additive scoring is mathematically sound. The dynamic programming solution with O(N²) complexity is efficient, and the theoretical analysis (including Corollary 3.1.1) is rigorous.
3. Using only start/end frames for interval representation (Eq. 3.5) is clever, avoiding variable-length representations while maintaining efficiency for nearest neighbor search.

=== Weaknesses ===

1. The method is only evaluated on RLBench in simulation. Other more challenging manipulation benchmarks (e.g., LIBERO) and real-world robotic manipulation often involve more complex visual variations, occlusions, and dynamics that may challenge the visual similarity-based retrieval approach.
2. RDD requires access to the full training dataset of the visuomotor policy to build the retrieval database. This may not always be feasible, especially for proprietary or large-scale policies.
3. The experiments show that performance degrades when the retrieval database is down-sampled to 25% of its original size. This suggests that a large and comprehensive database is necessary for good performance, which could pose a scalability challenge as the number of tasks and skills grows.

---

> ### Author Rebuttal · Authors · 2025-07-30
>
> First of all, we sincerely thank the reviewer for the insightful feedback. We are encouraged that the reviewer pointed out the major strengths of our paper: 1) the significance of our problem formulation; 2) the mathematical rigor of our solution; and 3) the efficiency of key design choices. We address the reviewer’s comments and questions point by point in the following.
>
> # Response to Weakness 1 & Question 1
>
> We thank the reviewer for this insightful question regarding RDD’s handling of out-of-distribution (OOD) demonstrations with novel sub-tasks. While the primary objective of RDD is to align the planner with the visuomotor policy’s existing capabilities, this is a critical aspect for real-world applicability. **In such OOD scenarios, the objective changes to: aligning sub-task intervals to both existing visuomotor sub-tasks and potentially new sub-tasks.**
>
> Thanks to the flexible design of RDD's scoring function $\\tilde{J}$, we do have a simple yet effective mechanism to gracefully handle these OOD scenarios and prevent catastrophic failures. Our approach involves two stages: OOD identification and OOD-aware decomposition.
>
> **OOD Identification**
>
> First, we must determine if a given demonstration is, in fact, OOD. We can quantify the novelty of a demonstration by calculating $\\Delta$, the average similarity score of the optimal partition $\\mathrm{P}$ found by the standard RDD algorithm:
>
>
> $$
> \\Delta=\\frac{1}{|\\mathrm{P}|}\\sum_{\\mathcal{I} \\in \\mathrm{P}} \\tilde{J}(\\mathcal{I}).
> $$
>
> A low value of *$\\Delta$* indicates that even the best possible decomposition has a low similarity to the sub-tasks in the training data, which signals an OOD demonstration. A simple threshold can be used to trigger our adaptive mechanism.
>
> **OOD-aware Decomposition**
>
> Once an OOD demonstration is identified, we introduce a modified similarity measure, $\\mathbf{sim}_{OOD}$, which incorporates a general interval scoring function $\\mathbf{G}$:
>
>
> $$
> \\mathbf{sim}_{OOD}(\\mathcal{I}_j^i,\\tilde{\\mathcal{I}_j^i})=\\mathbf{sim}(\\mathcal{I}_j^i,\\tilde{\\mathcal{I}_j^i})+\\beta \\mathbf{G}(\\mathcal{I}_j^i).
> $$
>
> Here, $\\mathbf{G}$ evaluates how well a proposed interval aligns with "general" sub-tasks, independent of the planner's specific training set. The hyperparameter $\\beta$ balances the trade-off between aligning with known sub-tasks (the original RDD objective) and discovering novel, generalizable sub-tasks. $\\mathbf{G}$ can be implemented using functions like UVD [A] to measure how well an interval conforms to generic change-point detection heuristics:
>
> $$
> \\mathbf{G}(\\mathcal{I})=-\\frac{1}{|\\mathcal{I}|}\\mathbf{abs}(b-\\mathbf{UVD}(e,\\mathcal{I})),
> $$
>
> where $b,e$ represent the index of the beginning and ending frame of interval $\\mathcal{I}$. $\\mathbf{UVD}(e,\\mathcal{I})$ gives the index of the UVD predicted beginning frame, given the goal frame on $e$.
>
> ## Additional Experimental Validation on LIBERO
>
> For OOD scenarios, we conducted additional experiments on LIBERO. To simulate a highly OOD scenario, we used the human-operated demonstration dataset from RoboCerebra [B] where demos are highly diverse in objects, task goals, and arrangements. We evaluated the quality of the decomposition against ground-truth segmentations using the mean Intersection-over-Union (mIoU) (see response to Reviewer DAU1 for details). We use 560 demos to build the RDD database and test on the remaining 140 demos.
>
> | Methods             | mIoU (LIBERO OOD Test) |
> | ------------------- | ----------------- |
> | UVD                 | 0.598             |
> | RDD ($\\beta=0.25$) | 0.624 ( ↑ 4.4%)   |
> | RDD ($\\beta=0.1$)  | 0.630 ( ↑ 5.3%)   |
> | RDD ($\\beta=0.05$) | 0.614 ( ↑ 2.6%)   |
>
> The results clearly show that our OOD-aware **RDD not only avoids catastrophic failure but also outperforms the baseline UVD decomposer in OOD settings.** RDD can successfully leverage even limited similarities in novel scenarios to produce a superior decomposition.
>
> ## Additional Experimental Validation on Real-World Dataset
>
> For in-distribution alignment (original goal of RDD), we test RDD on the real-world manipulation dataset AgiBotWorld-Alpha [C] to show the generalization ability of RDD visual features.  We test RDD and UVD on the task "supermarket" where 152 demos are used to build the RDD database and 37 demos are used for testing.
>
> | Methods | mIoU (AgiBotWorld Alignment Test) |
> | ------- | --------------------------- |
> | UVD     | 0.506                       |
> | RDD     | 0.706 ( ↑ 39.6%)            |
>
> RDD significantly outperforms the baseline method UVD, proving its generalization ability.
>
> These results demonstrate the robustness and adaptability of the RDD framework. We will add the above discussions to the revised paper.
>
> # Response to Weakness 2
>
> We thank the reviewer for the insightful feedback. We acknowledge that RDD requires access to the visuomotor policy’s training data, which could be a limitation for proprietary policies. However, we argue that the strong and growing trend in the robotics community toward open-sourcing models and datasets makes RDD increasingly relevant and applicable.
>
> For instance, the recent release of powerful, open-sourced models like OpenVLA is often trained on large-scale public datasets such as the Open X-Embodiment dataset. RDD’s core contribution is bridging the gap between high-level planners and low-level policies in hierarchical VLAs.
>
> As the community continues to embrace and build upon open-source resources, RDD is well-positioned to be a valuable tool for effectively leveraging these powerful models. We will clarify this application scope in the revised paper to highlight RDD’s utility in this emerging research landscape.
>
> # Response to Weakness 3
>
> We thank the reviewer for raising the important point about scalability. Since the offline vector dataset construction is performed only once, the core efficiency concern lies in the online decomposition process. Here, we highlight that the online decomposition process is efficient and scalable in two ways:
>
> **1. Linear Time Complexity**: As formally stated in **Corollary 3.1.1** in the paper, with a maximum sub-task duration (a mild assumption for real-world robotics), our dynamic programming-based solver has a linear time complexity of **O(N)** with respect to the number of demonstration frames. Our experimental results in Figure 3 confirm this linear growth.
>
> **2. Efficient Search & Parallelization**: The nearest neighbor search can be significantly accelerated using GPU-accelerated libraries like **FAISS [D]** or **Milvus [E]**. Furthermore, our decomposition algorithm is inherently parallelizable, allowing for substantial speed-ups through distributed computing.
>
>
> # Response to Question 2
>
> We thank the reviewer for their insightful question regarding RDD’s sensitivity to the quality of sub-task segmentation. We would like to address this by highlighting two key aspects of our approach:
>
> First, we wish to clarify RDD’s primary objective. The core contribution is not to learn new skills or correct flawed ones, but to enable a high-level planner to effectively utilize the skills that the low-level visuomotor policy already possesses. **Therefore, RDD is largely agnostic to the "optimality" of the skills themselves. This ensures the planner generates commands that the policy can reliably execute, rather than potentially "better" ones it cannot handle.**
>
> Second, in scenarios where the visuomotor policy's training data contains significant noisy samples that the policy fails to learn, **RDD can be easily integrated with dataset curation techniques [F, G].** These methods can serve as a pre-processing step to filter the visuomotor training set. For instance, CUPID [F] computes an "action influence" score for state-action pairs that can be used to evaluate each segment's contribution to the policy's final behavior. By applying a simple threshold, low-influence or flawed segments can be pruned from the dataset before RDD uses it as a reference. This would prevent catastrophic failures by ensuring RDD aligns demonstrations only with high-quality, influential sub-tasks.
>
> We will add these clarifying discussions to the revised version of our paper.
>
> # Response to Question 3
>
> We thank the reviewer for this insightful question. This issue is most apparent in sub-tasks with exceptionally long horizons (e.g., spanning several minutes, whereas typical sub-tasks in our experiments span tens of seconds). In such cases, the intermediate states and temporal evolution, which are not fully captured by our default start-end frame similarity, become crucial for disambiguating the sub-task’s semantic meaning.
>
> However, a key strength of our RDD framework is its modularity. The interval similarity measure, **sim**, is a plug-and-play component that offers a convenient and flexible interface to incorporate domain-specific knowledge or more powerful representations without altering the core dynamic programming algorithm. For instance, one could readily implement a more sophisticated **sim** function using powerful video encoders like VideoCLIP [H], which are designed to capture richer temporal and semantic information from video segments. Our novel, general problem formulation opens up exciting avenues for future research.
>
> # References
>
> [A] Zhang, et al. "Universal visual decomposer:" ICRA, 2024.
>
> [B] Han, et al. "RoboCerebra" arXiv preprint (2025).
>
> [C] Bu, et al. "Agibot world colosseo." arXiv preprint (2025).
>
> [D] Johnson, et al. "Billion-scale similarity search with GPUs." *IEEE Transactions on Big Data* 7.3 (2019).
>
> [E] Wang, et al. "Milvus: A purpose-built vector data management system." SIGMOD. 2021.
>
> [F] Agia, et al. "CUPID: Curating Data your Robot Loves with Influence Functions." *RSS.* 2025.
>
> [G] Hejna, et al. "Robot data curation with mutual information estimators." arXiv preprint (2025).
>
> [H] Xu, et al. "VideoCLIP" arXiv preprint (2021).

---

> > ### Comment · Reviewer_9Wsv · 2025-08-05
> >
> > I thank the authors for their significant effort in rebuttal and conducting extra experiments. The response largely addresses my concerns, although minor questions regarding scalability remain. Nevertheless, the paper's strengths outweigh its weaknesses. Given the growing trend in the robotics community toward open-sourcing models and datasets, as the authors noted, the proposed RDD framework is increasingly useful. I therefore maintain my initial rating of borderline accept.

---

> ### Author Response · Authors · 2025-08-06
> **Thank You for Championing Acceptance and Further Clarification**
>
> Dear Reviewer 9Wsv,
>
> Thank you for your consistent support and for championing the acceptance of our work!
>
> Regarding your question about scalability, **we respectfully emphasize that RDD is highly scalable.** The core is the **Linear Time Complexity of RDD.** As formally stated in **Corollary 3.1.1** in the paper, with a maximum sub-task duration (a mild assumption for real-world robotics), our dynamic programming-based solver has a linear time complexity of O(N) with respect to the number of demonstration frames. Our experimental results in Figure 3 confirm this linear growth.
>
> ## **Experimental Validation of RDD’s Scalability**
>
> The nearest neighbor (NN) search in RDD can be significantly accelerated using **GPU-accelerated libraries like FAISS [A] or Milvus [B].** Furthermore, **our decomposition algorithm is inherently parallelizable** (as described in Algorithm 1), allowing for substantial speed-ups through distributed computing.
>
> We conduct experiments on a typical database of **10 million entries** (**mainstream policy training dataset scale, as shown in the next section**) of **2048** dimensions (same dimension as our main experiment in Table 1, and the FAISS can achieve >300 NN queries per second on one NVIDIA 4090 GPU. Under this setting, RDD only needs < 2 minutes to decompose a 500-frame video, with a max interval length of 100 frames. (44549 NN queries in total). In other words, **as part of the offline dataset building process, RDD can decompose demonstrations at a high speed of 4.3 fps, a significant result shows high scalability of RDD.**
>
> | Hardware        | Vector Dim                 | Vector Num     | FAISS NN queries per second | Max Interval Frame Num | Demonstration Frame Num | RDD Time (second) |
> | --------------- | -------------------------- | -------------- | --------------------------- | ---------------------- | ----------------------- | ----------------- |
> | One NVIDIA 4090 | **2048 (same as Table 1)** | **10 million** | 386                         | 100                    | 500 (5 fps)             | **115 (4.3 fps)** |
>
> Besides, *RDD has a tolerance to 50% dataset down-sampling*. Table 7 (in the supplementary materials) shows that RDD can tolerate 50% down-sampling of the policy dataset with negligible performance drop. This result is significant and further highlights the efficiency and practicality of RDD.
>
> ## Scale of Mainstream Robotics Datasets
>
> **To support the aforementioned experiment settings**, here we provide the scale of some of the most popular open-sourced robotics datasets. Given the fact that these datasets have ~ 1 million trajectories (demonstrations), assuming each demonstration can be decomposed into 10 sub-tasks, **the mainstream policy training datasets typically have ~ 10 million sub-tasks. (~ 10 million entries in the database).**
>
> - ***The Open X-Embodiment (OXE) Dataset [C]:*** A landmark collaboration among 21 institutions, OXE provides over **1 million robot trajectories** from 22 different robot embodiments. Its explicit goal is to foster the development of generalist models, demonstrating that the community is actively removing the proprietary data barriers of the past. The explicit purpose of OXE is to provide a standardized, large-scale resource to train generalist models like RT-1-X and RT-2-X, which have demonstrated significant performance gains by training on this diverse data.
> - ***Hugging Face SmolVLA Dataset:*** The emergence of models like SmolVLA [D], a capable vision-language-action model trained entirely on **23k episodes** from 487 open-sourced community datasets through the LeRobot framework, outperforms the closed-source-dataset policy pi0 [G].
> - ***NVIDIA physical AI dataset:*** Major industry players are also championing this trend. For example, NVIDIA’s recent release of a massive, open-source physical AI dataset [E] ( initially 15 TB representing over **320,000 trajectories** ) with plans to become the “world’s largest unified and open dataset for physical AI.”
> - ***AgiBot World:*** Projects like AgiBot World [F] are providing not just datasets but complete open-source toolchains and standardized data collection pipelines, further enriching the public ecosystem. It has collected over **1 million trajectories** on over 100 homogeneous robots. Their proposed model GO-1, entirely trained on this open-sourced dataset, outperforms the closed-source dataset policy pi0 [G].
>
>
>
> [A] Johnson, et al. "Billion-scale similarity search with GPUs." *IEEE Transactions on Big Data* 7.3 (2019).
>
> [B] Wang, et al. "Milvus: A purpose-built vector data management system." SIGMOD. 2021.
>
> [C] O’Neill, et al. “Open x-embodiment” *2024 ICRA*. IEEE, 2024.
>
> [D] Shukor, et al. “Smolvla” *arXiv preprint* (2025).
>
> [E] Nvidia, “Physical AI - a nvidia Collection” Jul, 2025
>
> [F] Bu, et al. “Agibot world colosseo.” *arXiv preprint* (2025).
>
> [G] Black, et al. “$\\pi_0 $: A Vision-Language-Action Flow Model for General Robot Control.” *arXiv preprint* (2024).

---

### Note · Authors · 2025-08-12

**Dear Chairs and Reviewers,**

We sincerely thank you for your time and valuable feedback during the review process of our work!

We have addressed each reviewer’s comments and concerns in detail, and below we summarize the discussion and reaffirm our main contributions.

## Summary of Contributions

1. **Novel Problem**: We are the first to address a critical issue in VLA task planning: the high-level VLM planner’s lack of awareness of the low-level visuomotor policy’s capabilities, which can lead to sub-optimal sub-task instructions.
2. **Scalable & Adaptive Solution**: We propose a new problem formulation and algorithm with detailed theoretical analysis. We cast the visuomotor-aware planner fine-tuning dataset construction as an optimal partitioning problem and present an effective and efficient linear-time algorithm.
3. **Extensive Validation**: We conduct extensive experiments across both simulation and real-world benchmarks, including handling out-of-distribution novel sub-tasks.

## Summary of Comments and Responses

| Questions                                                    | From Reviewers | Details of Our Response                                      |
| ------------------------------------------------------------ | -------------- | ------------------------------------------------------------ |
| How RDD handles novel sub-tasks?  | 9Wsv, DAU1     | Rebuttal to Reviewer 9Wsv (30 Jul 2025, 12:52).              |
| Evaluation on additional benchmarks, including real-world datasets. | 9Wsv, wuNw     | Rebuttal to Reviewer 9Wsv (30 Jul 2025, 12:52).              |
| Open access to visuomotor policy's training set   | 9Wsv | Follow-up (05 Aug 2025, 18:26) & Rebuttal (30 Jul 2025, 12:52) to Reviewer 9Wsv |
| Scalability of RDD    | 9Wsv      | Follow-up (05 Aug 2025, 18:26) & Rebuttal (30 Jul 2025, 12:52) to Reviewer 9Wsv |
| Sensitivity of RDD to sub-optimal sub-tasks     | 9Wsv   | Rebuttal (30 Jul 2025, 12:52) to Reviewer 9Wsv   |
| Technical novelties | DAU1   | Rebuttal (30 Jul 2025, 13:07) to Reviewer DAU1     |
| Intermediate evaluation metrics  | DAU1   | Rebuttal (30 Jul 2025, 13:07) to Reviewer DAU1      |
| Significance of the improvement | wuNw   | Rebuttal (30 Jul 2025, 13:11) to Reviewer wuNw   |
| Compare RDD with large vision-language models | G7RJ  | Rebuttal (30 Jul 2025, 13:12) to Reviewer G7RJ  |

---

Thank you again for your time, support, and efforts in reviewing our work!

Warm regards,
The Authors
Paper #7445 – *"RDD"*

---

### Decision · Program_Chairs · 2025-09-17

**Decision:**

Accept (poster)

**Comment:**

This submission focuses on the alignment between high-level planners and low-level visuomotor policy for long-horizon robotic manipulation. The authors propose a Retrieval-based Demonstration Decomposer, which a a training-free method to  automatically decomposes demonstrations into sub-tasks. Experiment on RLBench demonstrate the performance under various settings. All reviewers vote for borderline acceptance.
AC recommend accept as poster.